# Stochastic Stein Discrepancies

**Jackson Gorham**
Whisper.ai, Inc
jackson@whisper.ai

**Anant Raj**
MPI for Intelligent Systems
Tübingen, Germany
anant.raj@tuebingen.mpg.de

**Lester Mackey**
Microsoft Research New England
lmackey@microsoft.com

## Abstract

Stein discrepancies (SDs) monitor convergence and non-convergence in approximate inference when exact integration and sampling are intractable. However, the computation of a Stein discrepancy can be prohibitive if the Stein operator – often a sum over likelihood terms or potentials – is expensive to evaluate. To address this deficiency, we show that *stochastic Stein discrepancies* (SSDs) based on subsampled approximations of the Stein operator inherit the convergence control properties of standard SDs with probability 1. Along the way, we establish the convergence of Stein variational gradient descent (SVGD) on unbounded domains, resolving an open question of Liu (2017). In our experiments with biased Markov chain Monte Carlo (MCMC) hyperparameter tuning, approximate MCMC sampler selection, and stochastic SVGD, SSDs deliver comparable inferences to standard SDs with orders of magnitude fewer likelihood evaluations.

## 1   Introduction

Markov chain Monte Carlo (MCMC) methods [7] provide asymptotically correct sample estimates $\frac{1}{n}\sum_{i=1}^{n} h(x_i)$ of the complex integrals $\mathbb{E}_P[h(Z)] = \int h(z)dP(z)$ that arise in Bayesian inference, maximum likelihood estimation [20], and probabilistic inference more broadly. However, MCMC methods often require cycling through a large dataset or a large set of factors to produce each new sample point $x_i$. To avoid this computational burden, many have turned to scalable approximate MCMC methods [e.g. 1, 8, 14, 39, 50], which mimic standard MCMC procedures while using only a small subsample of datapoints to generate each new sample point. These techniques reduce Monte Carlo variance by delivering larger sample sizes in less time but sacrifice asymptotic correctness by introducing a persistent bias. This bias creates new difficulties for sampler monitoring, selection, and hyperparameter tuning, as standard MCMC diagnostics, like trace plots and effective sample size, rely upon asymptotic exactness.

To effectively assess the quality of approximate MCMC outputs, a line of work [9, 21–23, 27, 35] developed computable *Stein discrepancies* (SDs) that quantify the maximum discrepancy between sample and target expectations and provably track sample convergence to the target $P$, even when explicit integration and direct sampling from $P$ are intractable. SDs have since been used to compare approximate MCMC procedures [2], test goodness of fit [11, 27, 28, 34], train generative models [40, 48], generate particle approximations [9, 10, 19], improve particle approximations [25, 32, 33], compress samples [42], conduct variational inference [41], and estimate parameters in intractable models [5].

However, the computation of the Stein discrepancy itself can be prohibitive if the Stein operator applied at each datapoint – often a sum over datapoint likelihoods or factors – is expensive to evaluate. This expense has led some users to heuristically approximate Stein discrepancies by subsampling data points [2, 33, 41]. In this paper, we formally justify this practice by proving that *stochastic Stein discrepancies* (SSDs) based on subsampling inherit the desirable convergence-tracking properties of standard SDs with probability 1. We then apply our techniques to analyze a scalable stochastic variant of the popular Stein variational gradient descent (SVGD) algorithm [33] for particle-based variational inference. Specifically, we generalize the compact-domain convergence results of Liu [31] to show, first, that SVGD converges on unbounded domains and, second, that stochastic SVGD (SSVGD) converges to the same limit as SVGD with probability 1. We complement these results with a series of experiments illustrating the application of SSDs to biased MCMC hyperparameter tuning, approximate MCMC sampler selection, and particle-based variational inference. In each case, we find that SSDs deliver inferences equivalent to or more accurate than standard SDs with orders of magnitude fewer datapoint accesses.

The remainder of the paper is organized as follows. In Section 2, we review standard desiderata and past approaches for measuring the quality of a sample approximation. In Section 3, we provide a formal definition of stochastic Stein discrepancies for scalable sample quality measurement and present a stochastic SVGD algorithm for scalable particle-based variational inference. We provide probability 1 convergence guarantees for SSDs and SSVGD in Section 4 and demonstrate their practical value in Section 5. We discuss our findings and posit directions for future work in Section 6.

**Notation**  For vector-valued $g$ on $\mathcal{X} \subseteq \mathbb{R}^d$, we define the expectation $\mu(g) \triangleq \int g(x) d\mu(x)$ for each probability measure $\mu$, the divergence $\langle \nabla, g(x) \rangle \triangleq \sum_{j=1}^d \frac{\partial}{\partial x_j} g_j(x)$, and the $\|\cdot\|_2$ boundedness and Lipschitzness parameters $\|g\|_\infty \triangleq \sup_{x \in \mathbb{R}^d} \|g(x)\|_2$ and $\mathrm{Lip}(g) \triangleq \sup_{x \neq y \in \mathcal{X}} \frac{\|g(x) - g(y)\|_2}{\|x - y\|_2}$. For any matrix $A$, let $\|A\|_{\mathrm{op}} \triangleq \sup_{\|x\|_2 \leq 1} \|Ax\|_2$ be the operator norm of $A$. For any $L \in \mathbb{N}$, we write $[L]$ for $\{1, \ldots, L\}$. We write $\Rightarrow$ for the weak convergence and $\overset{a.s.}{\Rightarrow}$ for almost sure convergence of probability measures. We denote the set of continuous functions and continuously differentiable functions on $\mathcal{X}$ as $C(\mathcal{X})$ and $C^1(\mathcal{X})$ respectively, and use the shorthand $C$ and $C^1$ whenever $\mathcal{X} = \mathbb{R}^d$. We also denote the set of functions on $\mathbb{R}^d \times \mathbb{R}^d$ continuously differentiable in both arguments by $C^{(1,1)}$.

## 2  Measuring Sample Quality

Consider a target distribution $P$ supported on $\mathcal{X} \subseteq \mathbb{R}^d$. We assume that exact expectations under $P$ are unavailable for many functions of interest, so we will an employ a discrete measure $Q_n \triangleq \frac{1}{n} \sum_{i=1}^n \delta_{x_i}$ based on a sample $(x_i)_{i=1}^n$ to approximate expectations under $P$. Importantly, we will make no assumptions on the origins or nature of the sample points $x_i$; they may be the output of i.i.d. sampling, drawn from an arbitrary Markov chain, or even generated by a deterministic quadrature rule.

To assess the usefulness of a given sample, we seek a quality measure that quantifies how well expectations under $Q_n$ match those under $P$. At the very least, this quality measure should *(i)* determine when $Q_n$ converges to the target $P$, *(ii)* determine when $Q_n$ does not converge to $P$, and *(iii)* be computationally tractable. *Integral probability metrics* (IPMs) [37] are natural candidates, as they measure the maximum absolute difference in expectation between probability measures $\mu$ and $\nu$ over a set of test functions $\mathcal{H}$:

$$d_{\mathcal{H}}(\mu, \nu) \triangleq \sup_{h \in \mathcal{H}} |\mathbb{E}_\mu[h(X)] - \mathbb{E}_\nu[h(Z)]|.$$

Moreover, for many IPMs, like the Wasserstein distance ($\mathcal{H} = \{h : \mathcal{X} \to \mathbb{R} \mid \mathrm{Lip}(h) \leq 1\}$) and the Dudley metric ($\mathcal{H} = \{h : \mathcal{X} \to \mathbb{R} \mid \|h\|_\infty + \mathrm{Lip}(h) \leq 1\}$), convergence of $d_{\mathcal{H}}(Q_n, P) \to 0$ implies that $Q_n \Rightarrow P$, in satisfaction of Desideratum (ii). Unfortunately, these same IPMs typically cannot be computed without exact integration under $P$. Gorham and Mackey [21] circumvented this issue by constructing a new family of IPMs – *Stein discrepancies* – from test functions known a priori to be mean zero under $P$. Their construction was inspired by Charles Stein's three-step method for proving central limit theorems [45]:

1. Identify an operator $\mathcal{T}$ that generates mean-zero functions on its domain $\mathcal{G}$:
$$\mathbb{E}_P[(\mathcal{T}g)(Z)] = 0 \text{ for any } g \in \mathcal{G}.$$

The chosen *Stein operator* $\mathcal{T}$ and *Stein set* $\mathcal{G}$ together yield an IPM-type measure which eschews explicit integration under $P$:

$$\mathcal{S}(\mu, \mathcal{T}, \mathcal{G}) \triangleq d_{\mathcal{T}\mathcal{G}}(\mu, P) = \sup_{g \in \mathcal{G}} |\mathbb{E}_\mu[(\mathcal{T}g)(X)] - \mathbb{E}_P[(\mathcal{T}g)(Z)]| = \sup_{g \in \mathcal{G}} |\mathbb{E}_\mu[(\mathcal{T}g)(X)]|. \quad (1)$$

Gorham and Mackey [21] named this measure the *Stein discrepancy*.

2. Lower bound the Stein discrepancy by an IPM known to dominate convergence in distribution. This is typically done for a large class of targets once and thus ensures that $\mathcal{S}(Q_n, \mathcal{T}, \mathcal{G}) \to 0$ implies $Q_n \Rightarrow P$ (Desideratum *(ii)*).

3. Upper bound the Stein discrepancy to ensure that the Stein discrepancy $\mathcal{S}(Q_n, \mathcal{T}, \mathcal{G}) \to 0$ when $Q_n$ converges suitably to $P$ (Desideratum *(i)*).

Prior work has instantiated a variety of Stein operators $\mathcal{T}$ and Stein sets $\mathcal{G}$ satisfying Desiderata *(i)-(iii)* for large classes of target distributions [9, 10, 18, 21–23, 27, 35, 45, 46]. We will focus on *decomposable operators*: $\mathcal{T} = \sum_{l=1}^{L} \mathcal{T}_l$ that decompose as a sum of $L$ base operators $\mathcal{T}_l$ that are less expensive to evaluate than $\mathcal{T}$. A prime example is the *Langevin Stein operator* derived in [21],

$$(\mathcal{T}_P g)(x) = \langle \nabla \log p(x), g(x) \rangle + \langle \nabla, g(x) \rangle, \quad (2)$$

applied to a differentiable posterior density $p(x) \propto \pi_0(x) \prod_{l=1}^{L} \pi(y_l|x)$ on $\mathbb{R}^d$ for $\pi_0$ a prior density, $\pi(\cdot|x)$ a likelihood function, and $(y_l)_{l=1}^{L}$ a sequence of observed datapoints. In this case, the Langevin operator $\mathcal{T}_P = \sum_{l=1}^{L} \mathcal{T}_l$ for

$$(\mathcal{T}_l g)(x) = \langle \nabla \log p_l(x), g(x) \rangle + \tfrac{1}{L} \langle \nabla, g(x) \rangle \quad \text{and} \quad p_l(x) \triangleq \pi_0(x)^{1/L} \pi(y_l|x), \quad (3)$$

so that each base operator involves accessing only a single datapoint.

## 3 Stochastic Stein Discrepancies

Whenever the Stein operator decomposes as $\mathcal{T} = \sum_{l=1}^{L} \mathcal{T}_l$, the standard Stein discrepancy (SD) objective (1) demands that every base operator $\mathcal{T}_l$ be evaluated at every sample point $x_i$; this cost can quickly become prohibitive if $L$ and $n$ are large. To alleviate this burden, we will consider a new class of discrepancy measures based on subsampling base operators. We emphasize that our aim in doing so is *not* to approximate standard SDs but rather to develop more practical alternative discrepancy measures that control convergence in their own right. To this end, we fix a batch size $m$ and, for each $i \in [n]$, independently select a uniformly random subset $\sigma_i$ of size $m$ from $[L]$. Then for any $\mathcal{G}$, we define the *stochastic Stein discrepancy* (SSD) as the random quantity

$$\mathcal{SS}(Q_n, \mathcal{T}, \mathcal{G}) \triangleq \sup_{g \in \mathcal{G}} \big| \tfrac{1}{n} \sum_{i=1}^{n} \tfrac{L}{m} (\mathcal{T}_{\sigma_i} g)(x_i) \big|, \quad (4)$$

where, for each $\sigma \subseteq [L]$, we introduce the subset operator $\mathcal{T}_\sigma \triangleq \sum_{l \in \sigma} \mathcal{T}_l$. In our running example of the Langevin posterior decomposition (3), we have

$$(\mathcal{T}_\sigma g)(x) = \langle \nabla \log p_\sigma(x), g(x) \rangle + \tfrac{m}{L} \langle \nabla, g(x) \rangle \quad \text{for} \quad p_\sigma(x) \triangleq \pi_0(x)^{m/L} \prod_{l \in \sigma} \pi(y_l|x),$$

so that each subset operator processes only a minibatch of $m$ datapoints.

By construction, the SSD reduces the number of base operator evaluations by a factor of $m/L$. Nevertheless, we will see in the Section 4 that SSDs inherit the convergence-determining properties of standard SDs with probability 1. Notably, the continued detection of convergence and non-convergence to $P$ is made possible by the use of an independent subset $\sigma_i$ per sample point. If, for example, the same minibatch of $m$ datapoints were used for all sample points instead, then the resulting discrepancy would determine convergence to an incorrect posterior conditioned on that minibatch rather than to the desired target $P$.

### 3.1 Stochastic kernel Stein discrepancies

Before turning to the convergence theory we pause to highlight a second property of practical import: when the Stein set is a unit ball of a reproducing kernel Hilbert space (RKHS), the SSD (9) admits a closed-form solution. We illustrate this for the Langevin operator (2) and the *kernel Stein set* [22]

$$\mathcal{G}_{k, \|\cdot\|} \triangleq \{ g = (g_1, \ldots, g_d) \mid \|v\|^* \le 1 \text{ for } v_j \triangleq \|g_j\|_{\mathcal{K}_k} \} \quad (5)$$

with arbitrary vector norm $\|\cdot\|$ and $\|\cdot\|_{\mathcal{K}_k}$ the RKHS norm of a reproducing kernel $k$ on $\mathbb{R}^d \times \mathbb{R}^d$.

---
**Algorithm 1** Stochastic Stein Variational Gradient Descent (SSVGD)
---
**Input:** Particles $(x_i^0)_{i=1}^n$, target $\nabla \log p = \sum_{l \in [L]} \nabla \log p_l$, kernel $k$, batch size $m$, rounds $R$

**for** $r = 0, \cdots, R-1$ **do**

    For each $j \in [n]$: sample independent mini-batch $\sigma_j$ of size $m$ from $[L]$

    For each $i \in [n]$: $x_i^{r+1} \leftarrow x_i^r + \epsilon_r \frac{1}{n} \sum_{j=1}^n \frac{L}{m} k(x_j^r, x_i^r) \nabla \log p_{\sigma_j}(x_j^r) + \nabla_{x_j^r} k(x_j^r, x_i^r)$

**Output:** Particle approximation $Q_{n,R}^m = \frac{1}{n} \sum_{i=1}^n \delta_{x_i^R}$ of the target $P$

---

**Proposition 1** (SKSD closed form). *If $k \in C^{(1,1)}$, then $\mathcal{SS}(Q_n, \mathcal{T}_P, \mathcal{G}_{k, \|\cdot\|}) = \|w\|$ where, $\forall j \in [d]$,*

$$w_j^2 \triangleq \frac{1}{n^2} \sum_{i=1}^n \sum_{i'=1}^n \left(\frac{L}{m} \nabla_{x_{ij}} \log p_{\sigma_i}(x_i) + \nabla_{x_{ij}}\right)\left(\frac{L}{m} \nabla_{x_{i'j}} \log p_{\sigma_{i'}}(x_{i'}) + \nabla_{x_{i'j}}\right) k(x_i, x_{i'}).$$

We call such discrepancies *stochastic kernel Stein discrepancies* (SKSDs) in homage to the standard kernel Stein discrepancies (KSDs) introduced in [11, 22, 34]. See App. A for the proof of Prop. 1.

**Related work**  Several research groups have stochastically approximated kernel-based Stein sets $\mathcal{G}$ to reduce the $\Omega(n^2)$ computational expense of goodness-of-fit testing [27, 28], measuring sample quality [27], and improving sample quality with Stein variational gradient descent [30] while leaving the original operator $\mathcal{T}$ unchanged. Others have improved communication efficiency by deploying standard SDs with special Stein sets featuring low-dimensional coordinate-dependent kernels [49, 53]. Here we focus on the distinct and complementary burden of evaluating an expensive Stein operator $\mathcal{T}$ at each sample point and note that the aforementioned approaches can be combined with datapoint subsampling to obtain substantial speed-ups. The recent, independent work of Hodgkinson et al. [25] uses a Langevin SKSD (in our terminology) to learn approximate importance sampling weights for an initial sample $Q_n$. Thm. 1 of [25] shows that the reweighted version of $Q_n$ asymptotically minimizes the associated KSD provided that the sample points $x_i$ are drawn from a $V$-uniformly ergodic Markov chain. In contrast, we offer convergence guarantees in Section 4 that apply to arbitrary sample points $x_i$, allow for non-kernel Stein discrepancies, and accommodate more general decomposable operators.

### 3.2  Stochastic Stein variational gradient descent

Our SSD analysis will also yield convergence guarantees for a stochastic version of the popular Stein variational gradient descent (SVGD) algorithm [33] on $\mathbb{R}^d$. SVGD iteratively improves a particle approximation $Q_n$ to a target distribution $P$ by moving each particle in the direction

$$g_{Q_n}^*(z) = \frac{1}{n} \sum_{j=1}^n k(x_j, z) \nabla \log p(x_j) + \nabla_{x_j} k(x_j, z)$$

that minimizes the KSD $\mathcal{S}(Q_n, \mathcal{T}_P, \mathcal{G}_{k, \|\cdot\|_2})$ with Langevin operator (2). However, when $\nabla \log p = \sum_{l=1}^L \nabla \log p_l$ is the sum of a large number of independently evaluated terms, each SVGD update can be prohibitively expensive. A natural alternative is to move each particle in the direction that minimizes the stochastic KSD $\mathcal{SS}(Q_n, \mathcal{T}_P, \mathcal{G}_{k, \|\cdot\|_2})$,

$$g_{Q_n, m}^*(z) = \frac{1}{n} \sum_{j=1}^n \frac{L}{m} k(x_j, z) \nabla \log p_{\sigma_j}(x_j) + \nabla_{x_j} k(x_j, z).$$

This amounts to replacing each $\nabla \log p(x_j)$ evaluation with an independent minibatch estimate on each update round to reduce the per-round gradient evaluation cost from $Ln$ to $mn$. The resulting *stochastic Stein variational gradient descent (SSVGD)* algorithm is detailed in Algorithm 1. Notably, after introducing SVGD, Liu and Wang [33] recommended subsampling as a heuristic approximation to speed up the algorithm. In Section 4.3, we aim to formally justify this practice.

## 4  Convergence Guarantees

In this section, we begin by showing that appropriately chosen SSDs detect the convergence and non-convergence of $Q_n$ to $P$ with probability 1 and end with new convergence results for SVGD and SSVGD. The former results will allow for an evolving sequence of Stein sets $(\mathcal{G}_n)_{n=1}^\infty$ to accommodate the graph Stein sets of [21, 23]. While we develop the most extensive theory for the popular Langevin Stein operator (2) with domain $\mathcal{X} = \mathbb{R}^d$, our results on detecting convergence

(Theorem 2), detecting bounded non-convergence (Theorem 4), and enforcing tightness (Prop. 5) apply to any decomposable Stein operator on any convex subset $\mathcal{X} \subseteq \mathbb{R}^d$. Throughout, we use the shorthand $\binom{[L]}{m} \triangleq \{\sigma \subseteq [L] : |\sigma| = m\}$ to indicate all subsets of $[L]$ of size $m$.

## 4.1 Detecting convergence with SSDs

We say that an SSD detects convergence if $\mathcal{SS}(Q_n, \mathcal{T}, \mathcal{G}_n) \rightarrow 0$ whenever $Q_n$ converges to $P$ in a standard probability metric, like the Wasserstein distance $W_a(Q_n, P) \triangleq \inf_{X \sim Q_n, Z \sim P} \mathbb{E}[\|X - Z\|_2^a]^{1/a}$ for $a \geq 1$. Our first result, proved in App. B, shows that an SSD detects Wasserstein convergence with probability 1 if its base operators $\mathcal{T}_\sigma$ generate continuous functions that grow no more quickly than a polynomial and have locally bounded derivatives. Theorem 2 is broad enough to cover all of the Stein operator-set pairings with SD convergence-detection results in [21–23].

**Theorem 2** (SSDs detect convergence). *Suppose that for some $a, c > 0$ and each $\sigma \in \binom{[L]}{m}$ and $n \geq 1$, $\mathcal{T}_\sigma \mathcal{G}_n \subset C(\mathcal{X})$, $\sup_{g \in \mathcal{G}_n} |(\mathcal{T}_\sigma g)(x)| \leq c(1 + \|x\|_2^a)$, $\sup_{n \geq 1, g \in \mathcal{G}_n, x, y \in K} \frac{|(\mathcal{T}_\sigma g)(x) - (\mathcal{T}_\sigma g)(y)|}{\|x-y\|_2} < \infty$ for each compact set $K$, and $P(\mathcal{T}g) = 0$ for all $g \in \mathcal{G}_n$. If $W_a(Q_n, P) \triangleq \inf_{X \sim Q_n, Z \sim P} \mathbb{E}[\|X - Z\|_2^a]^{1/a} \rightarrow 0$, then $\mathcal{SS}(Q_n, \mathcal{T}, \mathcal{G}_n) \xrightarrow{a.s.} 0$.*

## 4.2 Detecting non-convergence with SSDs

We say that an SSD detects non-convergence if $Q_n \not\Rightarrow P$ implies $\mathcal{SS}(Q_n, \mathcal{T}, \mathcal{G}_n) \not\rightarrow 0$. To establish this property, we first associate with every SSD, $\mathcal{SS}(Q_n, \mathcal{T}, \mathcal{G}_n)$, a *bounded Stein discrepancy*,

$$\mathcal{S}(Q_n, \mathcal{T}, \mathcal{G}_{b,n}) \rightarrow 0 \quad \text{with} \quad \mathcal{G}_{b,n} \triangleq \{g \in \mathcal{G}_n : \|\mathcal{T}_\sigma g\|_\infty \leq 1, \forall \sigma \in \binom{[L]}{m}\}, \tag{6}$$

in which each Stein function is constrained to be bounded under each subset operator $\mathcal{T}_\sigma$. We then show that SSDs detect non-convergence (culminating in Theorem 6) in a series of steps:

1. Theorem 3: If $Q_n \not\Rightarrow P$ then either a bounded SD $\not\rightarrow 0$ or $(Q_n)_{n=1}^\infty$ is not tight.
2. Theorem 4: If a bounded SD $\not\rightarrow 0$ then, with probability 1, its SSD $\not\rightarrow 0$.
3. Prop. 5: If $(Q_n)_{n=1}^\infty$ is not tight, then the SSD $\not\rightarrow 0$ surely.

We begin by showing that, for the popular Langevin operator (2) and each Stein set analyzed in [9, 10, 21–23], bounded SDs detect *tight* non-convergence. That is, if $Q_n \not\Rightarrow P$, then either $\mathcal{S}(Q_n, \mathcal{T}, \mathcal{G}_{b,n}) \not\rightarrow 0$ or some mass in the sequence $(Q_n)_{n=1}^\infty$ escapes to infinity. The proof is in App. C.

**Theorem 3** (Bounded SDs detect tight non-convergence). *Consider the Langevin Stein operator $\mathcal{T}_P$ (2) with Lipschitz $\nabla \log p$ satisfying* distant dissipativity *[16, 23] for some $\kappa > 0$ and $r \geq 0$:*

$$\langle \nabla \log p(x) - \nabla \log p(y), x - y \rangle \leq -\kappa \|x - y\|_2^2 + r, \quad \text{for all} \quad x, y \in \mathcal{X} = \mathbb{R}^d.$$

*Suppose $\sup_{x \in \mathcal{X}} \|\nabla \log p_\sigma(x)\|_2 / (1 + \|x\|_2) < \infty$ for each $\sigma \in \binom{[L]}{m}$, fix a sequence of probability measures $(Q_n)_{n=1}^\infty$, and consider the bounded Stein set $\mathcal{G}_{b,n}$ (6) for any of the following sets $\mathcal{G}_n$:*

*(A.1)* $\mathcal{G}_n = \mathcal{G}_{k,\|\cdot\|}$ *(5), the* kernel Stein set *of [22] with $k(x,y) = \Phi(x - y)$ for $\Phi \in C^2$ with non-vanishing Fourier transform.*

*(A.2)* $\mathcal{G}_n = \mathcal{G}_{\|\cdot\|} \triangleq \{g : \mathcal{X} \rightarrow \mathbb{R}^d \,|\, \sup_{x \neq y \in \mathcal{X}} \max(\|g(x)\|^*, \|\nabla g(x)\|^*, \frac{\|\nabla g(x) - \nabla g(y)\|^*}{\|x-y\|}) \leq 1\}$, *the* classical Stein set *of [21] with arbitrary vector norm $\|\cdot\|$.*

*(A.3)* $\mathcal{G}_n = \mathcal{G}_{\|\cdot\|, Q_n, G} \triangleq \{g \,|\, \forall x \in V, \, \max(\|g(x)\|^*, \|\nabla g(x)\|^*) \leq 1 \text{ and}, \, \forall (x,y) \in E,$
$\max(\frac{\|g(x) - g(y)\|^*}{\|x-y\|}, \frac{\|\nabla g(x) - \nabla g(y)\|^*}{\|x-y\|}, \frac{\|g(x) - g(y) - \nabla g(x)(x-y)\|^*}{\frac{1}{2}\|x-y\|^2}) \leq 1\}$, *the* graph Stein set *of [21] with arbitrary vector norm $\|\cdot\|$ and a finite graph $G = (V, E)$ with vertices $V \subset \mathcal{X}$.*

*If $Q_n \not\Rightarrow P$, then either $\mathcal{S}(Q_n, \mathcal{T}_P, \mathcal{G}_{b,n}) \not\rightarrow 0$ or $(Q_n)_{n=1}^\infty$ is not tight.*

Next, we prove in App. D that every SSD detects the non-convergence of its bounded SD.

**Theorem 4** (SSDs detect bounded SD non-convergence). *If $\mathcal{S}(Q_n, \mathcal{T}, \mathcal{G}_{b,n}) \not\to 0$, then, with probability 1, $\mathcal{SS}(Q_n, \mathcal{T}, \mathcal{G}_n) \not\to 0$.*

Finally, we prove in App. E that SSDs with coercive (radially unbounded) test functions *enforce tightness*, that is, remain bounded away from 0 whenever $(Q_n)_{n=1}^{\infty}$ is not tight.

**Proposition 5** (Coercive SSDs enforce tightness). *If $(Q_n)_{n=1}^{\infty}$ is not tight and $\frac{L}{m}\mathcal{T}_\sigma g$ is coercive and bounded below for some $g \in \bigcap_{n=1}^{\infty} \mathcal{G}_n$ and $\forall \sigma \in \binom{[L]}{m}$, then surely $\mathcal{SS}(Q_n, \mathcal{T}, \mathcal{G}_n) \not\to 0$.*

Taken together, these results imply that SSDs equipped with the Langevin operator and any of the convergence-determining Stein sets of [9, 10, 21–23] detect non-convergence with probability 1 under standard dissipativity and growth conditions on the subsampled operator.

**Theorem 6** (Coercive SSDs detect non-convergence). *Under the notation of Theorem 3, suppose $\nabla \log p$ is Lipschitz, $\sup_{x \in \mathcal{X}} \frac{\|\nabla \log p_\sigma(x)\|_2}{1 + \|x\|_2} < \infty$ for all $\sigma \in \binom{[L]}{m}$, and, for some $\kappa > 0$ and $r \geq 0$,*

$$\frac{L}{m}\langle \nabla \log p_\sigma(x) - \nabla \log p_\sigma(y), x - y \rangle \leq -\kappa\|x - y\|_2^2 + r, \quad \forall x, y \in \mathbb{R}^d \text{ and } \forall \sigma \in \binom{[L]}{m}.$$

*Consider the radial functions $\Phi_1(x) \triangleq (1 + \|x\|_2^2)^{\beta_1}$ for $\beta_1 \in (-1, 0)$ and $\Phi_2(x) \triangleq (\alpha + \log(1 + \|x\|_2^2))^{\beta_2}$ for $\alpha > 0, \beta_2 < 0$ underlying the inverse multiquadric and log inverse kernels [9] respectively. For each $n \geq 1$, suppose also that $\mathcal{G}_n$ satisfies (A.3), (A.2), or (A.1) with kernel $k(x, y) = \Phi_j(\Gamma(x - y))$ for $j \in \{1, 2\}$ and any positive definite matrix $\Gamma$. If $Q_n \not\Rightarrow P$, then, with probability 1, $\mathcal{SS}(Q_n, \mathcal{T}_P, \mathcal{G}_n) \not\to 0$.*

We prove this claim in App. F.

### 4.3 Convergence of SVGD and SSVGD

Discussing the convergence of SVGD and SSVGD on will require some additional notation. For each step size $\epsilon > 0$ and suitable probability measure $\mu$, define the SVGD update rule

$$T_{\mu,\epsilon}(x) = x + \epsilon\mathbb{E}_{X' \sim \mu}[\nabla \log p(X')k(X', x) + \nabla k(X', x)],$$

and let $\Phi_\epsilon(\mu)$ denote the distribution of $T_{\mu,\epsilon}(X)$ when $X \sim \mu$. If SVGD is initialized with the point set $(x_{i,0}^n)_{i=1}^n$, then the output of SVGD after each round $r$ is described by the recursion $Q_{n,r} = \Phi_{\epsilon_{r-1}}(Q_{n,r-1})$ for $r > 0$ with $Q_{n,0} \triangleq \frac{1}{n}\sum_{i=1}^n \delta_{x_{i,0}^n}$.

Liu [31] used this recursion to analyze the convergence of non-stochastic SVGD in three steps. First, Thm. 3.2 of [31] showed that, if the SVGD initialization $Q_{n,0}$ converges weakly to a probability measure $Q_{\infty,0}$ as $n \to \infty$, then, on each round $r > 0$, the $n$-point output $Q_{n,r}$ converges weakly to $Q_{\infty,r} \triangleq \Phi_{\epsilon_{r-1}}(Q_{\infty,r-1})$. Next, Thm. 3.3 of [31] showed that the Langevin KSD $\mathcal{S}(Q_{\infty,r}, \mathcal{T}_P, \mathcal{G}_k) \to 0$ as $r \to \infty$ for a suitable sequence of step sizes $\epsilon_r$. Finally, Thm. 8 of [22] implied that $Q_{\infty,r} \Rightarrow P$ for suitable kernels and targets $P$.

A gap in this analysis lies in the stringent assumptions of the first step: Thm. 3.2 of [31] only applies when $f(x, z) = \nabla \log p(x)k(x, z) + \nabla_x k(x, z)$ is both bounded and Lipschitz, but the growth of $\nabla \log p(x)k(x, z)$ typically invalidates both assumptions[1]. Indeed, Liu [31] remarks, "Therefore, the condition ... suggests that it can only be used when [the domain] is compact. It is an open question to establish results that can work for more general domain[s]." Our next theorem, proved in App. G, achieves this goal for $\mathbb{R}^d$ by showing that, on round $r$, both the SVGD output $Q_{n,r}$ and the SSVGD output $Q_{n,r}^m$ of Algorithm 1 converge to $Q_{\infty,r}$ with probability 1 under assumptions commonly satisfied by $p$ and $k$.

**Theorem 7** (Wasserstein convergence of SVGD and SSVGD). *Suppose SVGD and SSVGD are initialized with $Q_{n,0} = \frac{1}{n}\sum_{i=1}^n \delta_{x_{i,0}^n}$ satisfying $W_1(Q_{n,0}, Q_{\infty,0}) \to 0$. If for some $c_1, c_2 > 0$,*

$$\begin{aligned} \mathrm{Lip}(\nabla \log p(x)k(x, \cdot) + \nabla_x k(x, \cdot)) &\leq c_1(1 + \|x\|_2) \quad \text{and} \\ \mathrm{Lip}(\nabla \log p(\cdot)k(\cdot, z) + \nabla_x k(\cdot, z)) &\leq c_2(1 + \|z\|_2), \end{aligned} \tag{7}$$

*then* $W_1(Q_{n,r}, Q_{\infty,r}) \to 0$ *as* $n \to \infty$ *for each round* $r$. *If, in addition, for some* $c_0 > 0$,

$$\max_{\sigma \in \binom{[L]}{m}} \sup_{z \in \mathbb{R}^d} \|\nabla \log p_\sigma(x) k(x,z)\|_2 \leq c_0(1 + \|x\|_2), \tag{8}$$

$$\max_{\sigma \in \binom{[L]}{m}} \sup_{z \in \mathbb{R}^d} \|\nabla_x(\nabla \log p_\sigma(x) k(x,z))\|_{\mathrm{op}} \text{ is bounded on compact sets } K,$$

*then, for each round* $r$, $W_1(Q_{n,r}^m, Q_{n,r}) \overset{a.s.}{\to} 0$ *as* $n \to \infty$.

To illustrate the applicability of Theorem 7, we highlight that the growth assumptions (7) and (8) hold for standard bounded radial kernels like the Gaussian, Matérn, inverse multiquadric, and inverse log [9] kernels paired with any Lipschitz $\nabla \log p$ and linear-growth $\nabla \log p_\sigma$.

## 5 Experiments

In this section, we demonstrate the practical benefits of using SSDs as drop in replacements for standard SDs. In each of our experiments, the target is a posterior distribution of the form $p(x) \propto \prod_{l=1}^{L} p_l(x)$ where $p_l(x) \triangleq \pi_0(x)^{1/L} \pi(y_l|x)$ for $\pi_0$ a prior density, $\pi(\cdot|x)$ a likelihood function, and $(y_l)_{l=1}^{L}$ a sequence of observed datapoints. The SKSDs in Sections 5.1 and 5.2 use an inverse multiquadric base kernel $k(x,y) = (1 + \|x - y\|_2^2)^\beta$ with $\beta = -\frac{1}{2}$ as in [22]. Julia [6] code recreating the experiments in Sections 5.1 and 5.2 and Python code recreating the experiments in Section 5.3 is available at `https://github.com/jgorham/stochastic_stein_discrepancy`.

### 5.1 Hyperparameter selection for approximate MCMC

*Stochastic gradient Langevin dynamics* (SGLD) [50] with constant step size $\epsilon$ is an approximate MCMC method introduced as a scalable alternative to the popular Metropolis-adjusted Langevin algorithm [43]. A first step in using SGLD is selecting an appropriate step size $\epsilon$, as overly large values lead to severe distributional biases (see the right panel of the Fig. 1 triptych), while overly small values yield slow mixing (as in the left panel of the Fig. 1 triptych). In [50, Sec. 5.1], the posterior over the means of a Gaussian mixture model (GMM) was used to illustrate the utility of SGLD, and in [21, Sec. 5.3], the spanner graph Stein discrepancy was employed to select an appropriate $\epsilon > 0$ for a fixed computational budget. We recreate the experimental setup of [21, Sec. 5.3] to assess the ability of a stochastic KSD to effectively tune SGLD.

We used the same model parameterization as Welling and Teh [50], which was a posterior distribution with $L = 100$ likelihood terms contributing to the posterior density. We adopted the same experimental methodology as [21, Sec. 5.3]: for a range of $\epsilon$ values, we generated 50 independent SGLD pilot chains of length $n = 1000$. For each sample of size $n$, we computed the IMQ KSD without any subsampling and the SKSD with batch sizes $m = 1$ and $m = 10$. In Figure 1, we see that both SKSDs behave in step with the standard KSD: the choice of $\epsilon = 5 \times 10^{-3}$ minimizes the KSD over the average of the 50 trials for all variants of KSD. Moreover, the fastest SKSD required one hundredth the number of likelihood evaluations of the standard KSD. Hence, subsampling can lead to significant speed-ups with little degradation in inferential quality even when the total number of likelihood terms is moderate.

### 5.2 Selecting biased MCMC samplers

Gorham and Mackey [22, Sec. 4.4] used the KSD to choose between two biased sampling procedures. Namely, they compared two variants of the approximate MCMC algorithm *stochastic gradient Fisher scoring* (SGFS) [1]. The full variant of this sampler—called SGFS-f—requires inverting a $d \times d$ matrix to produce each sample iterate. A more computationally expedient variant—called SGFS-d—instead inverts that $d \times d$ matrix but first zeroes out all off-diagonal entries. Both MCMC samplers are uncorrected discretizations of a continuous-time process, and their invariant measures are asymptotically biased away from the target $P$. Accordingly, the SSD can be employed to assess whether the greater number of sample iterates generated by SGFS-d under a fixed computational budget outweighs the additional cost from asymptotic bias.

In both [22, Sec 4.4] and [1, Sec 5.1], the chosen target $P$ was a Bayesian logistic regression with a flat prior. The training set was constructed by selecting a subset of $10,000$ images from the MNIST dataset that had a 7 or 9 label, and then reducing each covariate vector of 784 pixel values

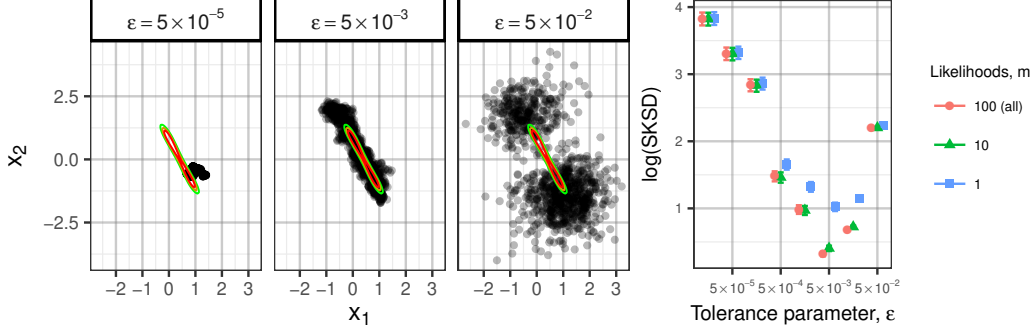

Figure 1: **Left:** Representative samples with $n = 1,000$ points obtained from SGLD with varying step sizes $\epsilon$. The contours represent high density regions of the bimodal posterior distribution. Notice the leftmost plot suffers from a lack of mixing, while the rightmost plot is far too overdispersed to fit the posterior. **Right:** For different subsampling sizes $m$ of the $L = 100$ likelihood terms contributing to the posterior, the mean IMQ SKSD ($\pm 1$ standard error) over 50 trials for each choice of $\epsilon$ is shown on a log scale. Even at extreme subsampling rates, the SKSD produces the same ranking of candidates and selects the same $\epsilon$ as the exact KSD.

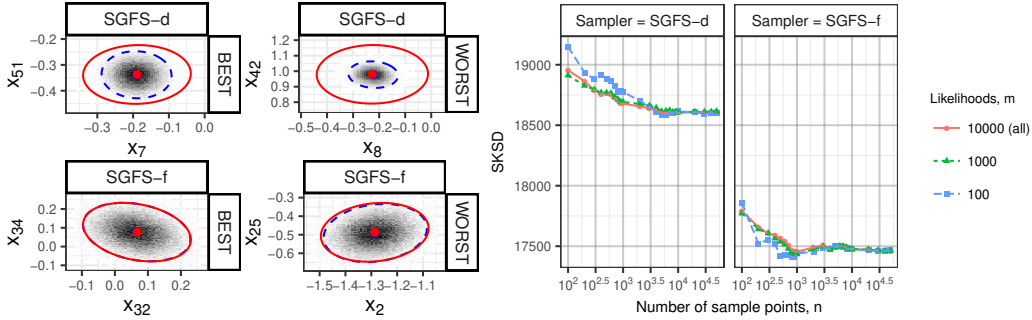

Figure 2: **Left:** The best and worst bivariate density plots of $50,000$ SGFS sample iterates that approximate the true target posterior distribution $P$. The overlaid lines are the bivariate marginal means and 95% confidence ellipses; the dashed blue are derived from SGFS samples and the solid red is derived from a surrogate ground truth sample. **Right:** Plot of exact KSD (red) and stochastic KSDs (green and blue) for each SGFS sampler vs. the number of sample iterates $n$.

to a dimension 50 vector via random projections. After including an intercept term, Ahn et al. [1] generated a posterior sample of $50,000$ sample iterates (each in $\mathbb{R}^{51}$) for both samplers. In [22, Sec 4.4], the authors showed the KSD preferred the sample iterates generated from SGFS-f for any number of sample iterates, while in [1, Sec 5.1], the authors showed even the best bivariate marginals generated by SGFS-d were inferior to SGFS-f at matching the target posterior $P$.

In Figure 2, we compare the exact KSDs with the stochastic KSDs obtained from sampling 100 and $1,000$ of the $10,000$ likelihood terms i.i.d. for each posterior sample iterate. Notice that the stochastic KSD prefers SGFS-f over SGFS-d for each subsampling parameter as well, in accordance with the exact KSD. However, the most aggressively subsampled stochastic KSD requires 100 times fewer likelihood evaluations than its standard analogue.

### 5.3 Particle-based variational inference with SSVGD

SVGD was developed to iteratively improve a $n$-point particle approximation $Q_n$ to a given target distribution. To illustrate the practical benefit of the stochastic SVGD algorithm analyzed in Section 4.3 over standard SVGD, we reproduce the Bayesian neural network experiment from [33, Sec. 5] on three datasets used in their experiment. We adopt the exact experimental setup of [33] and adapt their code to compare SSVGD (Algorithm 1) with minibatch sizes $m = 0.1L$ and $m = 0.25L$ with standard SVGD ($m = L$). The procedure was run 20 times for each configuration, and each time we started with an independently sampled train-test split. The training sets for the `boston`,

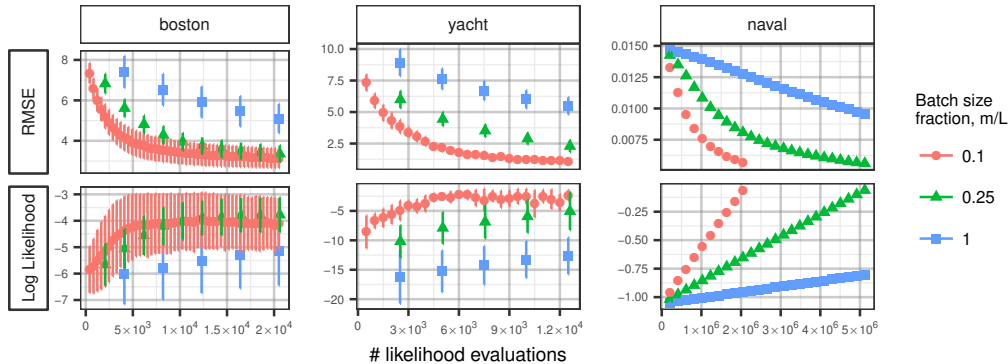

Figure 3: We plot the test RMSE and log likelihood for a Bayesian neural net as we approximate the posterior using stochastic SVGD over a set of sampling rates. The results shown above are the mean RMSE and log likelihood with $\pm 2$ standard errors of that mean across 20 train test splits for the boston, yacht and naval datasets. We see that for each likelihood budget considered the lower batch sizes produce more accurate approximations than full batch SVGD.

yacht, and naval datasets had $409, 209$, and $10, 241$ datapoints and $d = 13, 6$, and $17$ covariates, respectively. The boston dataset was first published in [24] while the latter two are available on the UCI repository [13]. The root mean-squared error (RMSE) and log likelihood were computed on the test set, and a summary is presented in Fig. 3. SSVGD yields more accurate approximations for all likelihood computation budgets considered, even for the modestly sized datasets, and this effect is magnified in the larger naval dataset.

## 6 Discussion and Future Work

To reduce the cost of assessing and improving sample quality, we introduced stochastic Stein discrepancies which inherit the convergence-determining properties of standard SDs with probability 1 while requiring orders of magnitude fewer likelihood evaluations. While our work was focused on measuring sample quality, we believe that other inferential tasks based on decomposable Stein operators can benefit from these developments. Prime candidates include SD-based goodness-of-fit testing [11, 27, 28, 34], KSD-based sampling [9, 10, 19], improving Monte Carlo estimation with control variates [3, 36, 38], improving sample quality through reweighting [25, 32] or thinning [42], and parameter estimation in intractable models [5]. Integrating variance reduction techniques [e.g., 12, 44] into the SSD computation is another promising direction, as the result could more closely mimic standard SDs while offering comparable computational savings. Finally, while the Langevin operator received special attention in our analysis, our results also extend to other popular Stein operators like the diffusion operators of [23] and the discrete operators of [52].

## Broader Impact

This work provides both producers and consumers of approximate inference techniques with a valid diagnostic for assessing those approximations at scale. It also analyzes a scalable algorithm (SSVGD) for improving approximate inference. We expect that many existing users of Stein discrepancies will want to employ stochastic Stein discrepancies to reduce their overall computational costs. In addition, the ready availability of a scalable diagnostic may stimulate the more widespread use of approximate MCMC methods. However, any inferential tool combined with the wrong data or inappropriate model can lead to incorrect and harmful conclusions, so care must be taken in interpreting the results of any downstream analysis.

## Acknowledgments and Disclosure of Funding

We thank Sungjin Ahn, Anoop Korattikara, and Max Welling for sharing their MNIST posterior samples. Part of this work was completed while Anant Raj was an intern at Microsoft Research.

## Footnotes

[1]Consider, for example, the standard Gaussian $\nabla \log p(x) = -x$ with any translation invariant kernel $k$ on $\mathbb{R}^d$.

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
