[Supplementary Material]

# A    Proof of Prop. 1: SKSD closed form

Our proof will parallel that of Gorham and Mackey [22, Prop. 2] for non-stochastic KSDs. For each $j \in [d]$ and each $\sigma_i$, we define the coordinate operators

$$\tfrac{L}{m}(\mathcal{T}_{\sigma_i}^j f)(x) \triangleq (\tfrac{L}{m}\nabla_{x_j}\log p_{\sigma_i}(x) + \nabla_{x_j})f(x)$$

for $f : \mathbb{R}^d \to \mathbb{R}$. For each $g = (g_1, \dots, g_d) \in \mathcal{G}_{k,\|\cdot\|}$ and $x \in \mathbb{R}^d$, our $C^{(1,1)}$ assumption on $k$ and the proof of [47, Cor. 4.36] imply that

$$(\mathcal{T}_{\sigma_i}g)(x) = \sum_{j=1}^d (\mathcal{T}_{\sigma_i}^j g_j)(x) = \sum_{j=1}^d \mathcal{T}_{\sigma_i}^j \langle g_j, k(x,\cdot)\rangle_{\mathcal{K}_k} = \sum_{j=1}^d \langle g_j, \mathcal{T}_{\sigma_i}^j k(x,\cdot)\rangle_{\mathcal{K}_k}.$$

Meanwhile, the result [47, Lem. 4.34] yields

$$\langle \tfrac{L}{m}\mathcal{T}_{\sigma_i}^j k(x_i,\cdot), \tfrac{L}{m}\mathcal{T}_{\sigma_{i'}}^j k(x_{i'},\cdot)\rangle = (\tfrac{L}{m}\nabla_{x_{ij}}\log p_{\sigma_i}(x_i) + \nabla_{x_{ij}})(\tfrac{L}{m}\nabla_{x_{i'j}}\log p_{\sigma_{i'}}(x_{i'}) + \nabla_{x_{i'j}})k(x_i, x_{i'})$$

for all $i, i' \in [n]$ and $j \in [d]$. Therefore, the advertised

$$w_j^2 = \tfrac{1}{n^2}\sum_{i=1}^n\sum_{i'=1}^n\langle \tfrac{L}{m}\mathcal{T}_{\sigma_i}^j k(x_i,\cdot), \tfrac{L}{m}\mathcal{T}_{\sigma_{i'}}^j k(x_{i'},\cdot)\rangle = \|\tfrac{1}{n}\sum_{i=1}^n \tfrac{L}{m}\mathcal{T}_{\sigma_i}^j k(x_i,\cdot)\|_{\mathcal{K}_k}^2.$$

Finally, our assembled results and norm duality give

$$\begin{aligned}
\mathcal{SS}(Q_n, \mathcal{T}_P, \mathcal{G}_{k,\|\cdot\|}) &= \sup_{g \in \mathcal{G}_{k,\|\cdot\|}} \sum_{j=1}^d \tfrac{1}{n}\sum_{i=1}^n \tfrac{L}{m}(\mathcal{T}_{\sigma_i}^j g_j)(x_i)\\
&= \sup_{\|g_j\|_{\mathcal{K}_k}=v_j, \|v\|^* \le 1} \sum_{j=1}^d \langle g_j, \tfrac{1}{n}\sum_{i=1}^n \tfrac{L}{m}\mathcal{T}_{\sigma_i}^j k(x_i,\cdot)\rangle_{\mathcal{K}_k}\\
&= \sup_{\|v\|^* \le 1} \sum_{j=1}^d v_j \|\tfrac{1}{n}\sum_{i=1}^n \tfrac{L}{m}[\mathcal{T}_{\sigma_i}^j k(x_i,\cdot)]\|_{\mathcal{K}_k}\\
&= \sup_{\|v\|^* \le 1} \sum_{j=1}^d v_j w_j = \|w\|.
\end{aligned}$$

# B    Proof of Theorem 2: SSDs detect convergence

We will find it useful to write

$$\mathcal{SS}(Q_n, \mathcal{T}, \mathcal{G}) = \sup_{g \in \mathcal{G}}\left|\tfrac{1}{n}\sum_{i=1}^n \tfrac{L}{m}\sum_{\sigma \in \binom{[L]}{m}} B_{i\sigma}(\mathcal{T}_\sigma g)(x_i)\right| \quad \text{for} \quad B_{i\sigma} \triangleq \mathbb{I}[\sigma = \sigma_i] \qquad (9)$$

$$= \sup_{g \in \mathcal{G}}\left|\binom{L}{m}^{-1}\sum_{\sigma \in \binom{[L]}{m}} \mu_{n\sigma}(\mathcal{T}_\sigma g)\right| \quad \text{for} \quad \mu_{n\sigma} \triangleq \binom{L}{m}\tfrac{L}{m}\tfrac{1}{n}\sum_{i=1}^n B_{i\sigma}\delta_{x_i}.$$

We will also write $BL_{\|\cdot\|} \triangleq \{h : \mathbb{R}^d \to \mathbb{R} : \|h\|_\infty + \text{Lip}(h) \le 1\}$ as the unit ball in the bounded Lipschitz metric, and for any $R > 0$, $B_R \triangleq \{x \in \mathbb{R}^d : \|x\|_2 \le R\}$ as the radius $R$ ball centered at the origin. For any set $K$, let $I_K(x) = \mathbb{I}[x \in K]$.

Our proof relies on a lemma, proved in App. B.1, that boosts almost sure convergence in distribution into almost sure uniform convergence for the expectations of all continuous functions dominated by a uniformly integrable, locally bounded $|f_0|$ with derivatives dominated by a locally bounded $|f_1|$.

**Lemma 8** (Convergence of random measures). *Consider two sequences of random measures $(\nu_n)_{n=1}^\infty$ and $(\tilde{\nu}_n)_{n=1}^\infty$ on $\mathbb{R}^d$, and suppose there exists an $R > 0$ such that $\nu_n(hI_{B_R}) - \tilde{\nu}_n(hI_{B_R}) \overset{a.s.}{\to} 0$ for each bounded and continuous $h$. Then, for $\mathcal{H} = BL_{\|\cdot\|}$,*

$$\sup_{h \in \mathcal{H}} |\nu_n(hI_{B_R}) - \tilde{\nu}_n(hI_{B_R})| \overset{a.s.}{\to} 0. \qquad (10)$$

*Suppose, in addition, that for every $S > 0$ there exists an $R \ge S$ such that (10) holds. Then if $f_0$ is almost surely uniformly $\nu_n$-integrable and uniformly $\tilde{\nu}_n$-integrable, and $f_0, f_1$ are bounded on each compact set, we have*

$$\sup_{h \in \mathcal{H}_f} |\nu_n(h) - \tilde{\nu}_n(h)| \overset{a.s.}{\to} 0,$$

*where $\mathcal{H}_f \triangleq \{h \in C(\mathbb{R}^d) : |h(x)| \le |f_0(x)|, \frac{|h(x)-h(y)|}{\|x-y\|_2} \le |f_1(x)| + |f_1(y)| \text{ for all } x, y \in \mathbb{R}^d\}$.*

Since $W_a(Q_n, P) \to 0$, [17, Proof of Cor. 1] implies that $Q_n(h) \to P(h)$ for all bounded continuous $h$ and that $f_0(x) = c(1 + \|x\|_2^a)$ is uniformly $Q_n$-integrable and $P$-integrable. Moreover, for each $\sigma \in \binom{[L]}{m}$, $\mu_{n\sigma}(h) - \frac{L}{m}Q_n(h) \overset{a.s.}{\to} 0$ for all bounded $h$ by Lemma 10, and thus $\mu_{n\sigma}(hI_{B_R}) - Q_n(hI_{B_R}) \overset{a.s.}{\to} 0$ for all bounded $h \in C(\mathbb{R}^d)$ and any $R > 0$. Since, for any compact set $K$, $\mu_{n\sigma}(|f_0|I_{K^c}) \leq \binom{L}{m}\frac{L}{m}Q_n(|f_0|I_{K^c})$, $f_0$ is also uniformly $\mu_{n\sigma}$-integrable. By assumption $f_1(x) = \omega(\|x\|_2)$ for $\omega(R) \triangleq \sup_n \sup_{g \in \mathcal{G}_n, x, y \in B_{2R}} \frac{|(\mathcal{T}_\sigma g)(x) - (\mathcal{T}_\sigma g)(y)|}{\|x - y\|_2}$ is bounded on any compact set.

Moreover, since $P$ is a finite measure, there are at most countably many values $R$ for which $P(\{x : \|x\|_2 = R\}) > 0$. Hence, for any $S > 0$ we can choose $R \geq S$ such that $B_R$ is a continuity set under $P$. For any such $R$, $Q_n(hI_{B_R}) - P(hI_{B_R}) \to 0$ for any bounded $h \in C(\mathbb{R}^d)$ by the Portmanteau theorem [29, Thm. 13.16], since $W_a(Q_n, P) \to 0$ implies convergence in distribution.

Finally, the assumption $P(\mathcal{T}g) = 0$ for all $g \in \mathcal{G}_n$, the triangle inequality, the continuity and polynomial growth of each function in $\mathcal{T}_\sigma \mathcal{G}_n$, and Lemma 8 applied first to $\mu_{n\sigma}$ and $(Q_n)_{n=1}^\infty$ for each $\sigma$ and then to $(Q_n)_{n=1}^\infty$ and $P$ together yield

$$\mathcal{SS}(Q_n, \mathcal{T}, \mathcal{G}_n) = \sup_{g \in \mathcal{G}_n} \left| \binom{L}{m}^{-1} \sum_{\sigma \in \binom{[L]}{m}} \mu_{n\sigma}(\mathcal{T}_\sigma g) - \frac{L}{m}Q_n(\mathcal{T}_\sigma g) + \frac{L}{m}Q_n(\mathcal{T}_\sigma g) - \frac{L}{m}P(\mathcal{T}_\sigma g) \right|$$

$$\leq \binom{L}{m}^{-1} \sum_{\sigma \in \binom{[L]}{m}} \sup_{h \in \mathcal{H}_f} |\mu_{n\sigma}(h) - \frac{L}{m}Q_n(h)| + \frac{L}{m}|Q_n(h) - P(h)| \overset{a.s.}{\to} 0.$$

### B.1 Proof of Lemma 8: Convergence of random measures

Fix any $R, \epsilon > 0$ and let $K = B_R$. By the Arzelà–Ascoli theorem [15, Thm. 8.10.6], there exists a finite $\epsilon/2$-subcover of the set of $K$-restrictions $\{h|_K : h \in \mathcal{H}\}$. Since any bounded continuous function on $K$ can be extended to a bounded continuous function on $\mathbb{R}^d$, there therefore exists a sequence of bounded continuous functions $(h_k)_{k=1}^m$ on $\mathbb{R}^d$ such that

$$\mathbb{P}(\sup_{h \in \mathcal{H}} |\nu_n(hI_K) - \tilde{\nu}_n(hI_K)| > \epsilon \text{ i.o.}) \leq \mathbb{P}(\max_{1 \leq k \leq m} |\nu_n(h_kI_K) - \tilde{\nu}_n(h_kI_K)| > \epsilon/2 \text{ i.o.})$$
$$\leq \sum_{k=1}^m \mathbb{P}(|\nu_n(h_k) - \tilde{\nu}_n(h_k)| > \epsilon/2 \text{ i.o.}) = 0,$$

where we have used the union bound and our almost sure convergence assumption for bounded continuous functions. The first result (10) now follows since $\epsilon$ was arbitrary.

We next assume that the event $\mathcal{E}$ on which $f_0$ is uniformly $\nu_n$ and $\tilde{\nu}_n$-integrable occurs with probability 1, and fix any $\epsilon > 0$. On $\mathcal{E}$ there exists $R_\epsilon > 0$ such that (10) holds and $\sup_n \max(\nu_n(|f_0|I_{K_\epsilon^c}), \tilde{\nu}_n(|f_0|I_{K_\epsilon^c})) \leq \epsilon/2$ for $K_\epsilon \triangleq B_{R_\epsilon}$. Furthermore, on $\mathcal{E}$,

$$\sup_{h \in \mathcal{H}_f} |\nu_n(h) - \nu_n(hI_{K_\epsilon})| + |\tilde{\nu}_n(h) - \tilde{\nu}_n(hI_{K_\epsilon})| \leq \sup_{h \in \mathcal{H}_f} \nu_n(|h|I_{K_\epsilon^c}) + \tilde{\nu}_n(|h|I_{K_\epsilon^c})$$
$$\leq \nu_n(|f_0|I_{K_\epsilon^c}) + \tilde{\nu}_n(|f_0|I_{K_\epsilon^c}) \leq \epsilon.$$

Therefore, the triangle inequality, fact that for each $R > 0$ there is a constant $c_R > 0$ such that $\{hI_{B_R} : h \in \mathcal{H}_f\} \subseteq \{c_R hI_{B_R} : h \in \mathcal{H}\}$, and our first result (10) give

$$\mathbb{P}(\sup_{h \in \mathcal{H}_f} |\nu_n(h) - \tilde{\nu}_n(h)| > 2\epsilon \text{ i.o.}) \leq \mathbb{P}(\mathcal{E}^c) + \mathbb{P}(\sup_{h \in \mathcal{H}_f} |\nu_n(hI_{K_\epsilon}) - \tilde{\nu}_n(hI_{K_\epsilon})| > \epsilon \text{ i.o.})$$
$$\leq \mathbb{P}(\mathcal{E}^c) + \mathbb{P}(c_{R_\epsilon} \sup_{h \in \mathcal{H}} |\nu_n(hI_{K_\epsilon}) - \tilde{\nu}_n(hI_{K_\epsilon})| > \epsilon \text{ i.o.})$$
$$= 0.$$

The second result now follows since $\epsilon$ was arbitrary.

## C  Proof of Theorem 3: Bounded SDs detect tight non-convergence

We consider each Stein set candidate in turn.

### C.1  Kernel Stein set

Suppose $\mathcal{G}_n$ satisfies (A.1). Since, for any vector norm $\|\cdot\|$ on $\mathbb{R}^d$, there exists $c_d$ such that $\{g \in \mathcal{G}_{k,\|\cdot\|_2} : \max_{\sigma \in \binom{[L]}{m}} \|\mathcal{T}_\sigma g\|_\infty \leq 1\} \subseteq c_d\{g \in \mathcal{G}_{k,\|\cdot\|} : \max_{\sigma \in \binom{[L]}{m}} \|\mathcal{T}_\sigma g\|_\infty \leq 1\}$ [4], it suffices to assume $\|\cdot\| = \|\cdot\|_2$.

**Choosing a convergence-determining IPM** $d_{\mathcal{H}}$    Consider the test function set $\mathcal{H}$ from [22, Sec E.1, Proof of Thm. 5] which satisfies

1. $\|h\|_\infty \leq 1$ and $\mathrm{Lip}(h) \leq 1 + \sqrt{d-1}$ for all $h \in \mathcal{H}$ and
2. $Q_n \not\Rightarrow P$ implies $d_{\mathcal{H}}(Q_n, P) \not\to 0$ for any sequence of probability measures $(Q_n)_{n \geq 1}$.

**Solving the Stein equation** $\mathcal{T}_P g_h = h - P(h)$    Let us define $\Xi(x) \triangleq (1 + \|x\|_2^2)^{1/2}$. By [22, Sec E.1, Proof of Thm. 5], for each $h \in \mathcal{H}$ there exists an accompanying function $g_h$ such that $\mathcal{T}_P g_h = h - P(h)$ and $\|\Xi g_h\|_\infty \leq \mathcal{M}_P$ for a constant $\mathcal{M}_P > 0$ independent of $h$.

**Smoothing the Stein function** $g_h$    Fix any $\rho \in (0,1]$, and let $U \sim \mathcal{N}(0, I)$. Since $\nabla \log p$ is Lipschitz, the argument in [22, Proof of Thm. 13] constructs a smoothed approximation $g_{h,\rho}(x) = \mathbb{E}[g_h(x - \rho U)]$ satisfying

$$\|\mathcal{T}_P g_{h,\rho} - \mathcal{T}_P g_h\|_\infty \leq C_1 \rho \tag{11}$$

for a constant $C_1$ independent of $h$ and $\rho$. Moreover, the following lemma shows that

$$\|\Xi g_{h,\rho}\|_\infty \leq \|\Xi g_h\|_\infty \sqrt{2}\mathbb{E}[1 + \|U\|_2] \leq \mathcal{M}_P' \triangleq \sqrt{2}\mathcal{M}_P(1 + \sqrt{d}),$$

where $\mathcal{M}_P$ is notably independent of $\rho$ and $h$.

**Lemma 9** (Smoothing preserves decay). *For each $g : \mathbb{R}^d \to \mathbb{R}^d$, $\epsilon \in [0,1]$, and absolutely integrable random vector $Y \in \mathbb{R}^d$,*

$$\sup_{x \in \mathbb{R}^d} \mathbb{E}[A(x)\|g(x - \epsilon Y)\|_2] \leq \sqrt{2}\|\Xi g\|_\infty \mathbb{E}[A(Y)] \quad for \quad A(x) \triangleq 1 + \|x\|_2. \tag{12}$$

**Proof**    For $B(y) \triangleq \sup_{x, u \in (0,1]} A(x)/\Xi(x - uy)$, we have

$$\sup_{x \in \mathbb{R}^d} \mathbb{E}[(1 + \|x\|_2)\|g(x - \epsilon Y)\|_2] = \sup_{x \in \mathbb{R}^d} \mathbb{E}\left[\frac{(1+\|x\|_2)}{\Xi(x-\epsilon Y)}\Xi(x - \epsilon Y)\|g(x - \epsilon Y)\|_2\right]$$

$$\leq \sup_{x \in \mathbb{R}^d} \|\Xi g\|_\infty \mathbb{E}\left[\frac{(1+\|x\|_2)}{\Xi(x-\epsilon Y)}\right] \leq \|\Xi g\|_\infty \mathbb{E}[B(Y)].$$

Moreover, $\Xi(z) \geq 2^{-1/2}(1 + \|z\|_2)$ for all $z$ implies that, for any $y$,

$$B(y) = \sup_{x, u \in (0,1]} \tfrac{A(x)}{\Xi(x-uy)} \leq \sup_{x, u \in (0,1]} \sqrt{2}\tfrac{A(x)}{1+\|x-uy\|_2} = \sup_{z, u \in (0,1]} \sqrt{2}\tfrac{A(z+uy)}{1+\|z\|_2}$$

$$\leq \sup_{z, u \in (0,1]} \sqrt{2}\tfrac{A(z)+u\|y\|_2}{1+\|z\|_2} \leq \sqrt{2}A(y),$$

where we used the triangle inequality in the penultimate inequality.    $\square$

**Truncating the smoothed Stein function** $g_{h,\rho}$    Fix any $\epsilon \in (0,1)$, and, since $(Q_n)_{n=1}^\infty$ is tight, select a compact set $K_\epsilon$ satisfying $\sup_n Q_n(K_\epsilon^c) \leq \epsilon$. The argument in [22, Proof of Thm. 13] identifies a truncation $g_{h,\rho,\epsilon}$ and a constant $C_0$ independent of $h$, $\epsilon$, and $\rho \in (0,1]$ such that, for all $x \in \mathbb{R}^d$,

$$\|g_{h,\rho,\epsilon}(x)\|_2 \leq \|g_{h,\rho}(x)\|_2 \quad \text{and}$$
$$|(\mathcal{T}_P g_{h,\rho,\epsilon})(x) - (\mathcal{T}_P g_{h,\rho})(x)| \leq C_0 \mathbb{I}[x \in K_\epsilon^c]. \tag{13}$$

Hence, $\|\Xi g_{h,\rho,\epsilon}\|_\infty \leq \|\Xi g_{h,\rho}\|_\infty \leq \mathcal{M}_P'$.

**Smoothing the truncation** $g_{h,\rho,\epsilon}$    By assumption, for all $\sigma \in \binom{[L]}{m}$, there is a constant $\beta > 0$ such that $\|\nabla \log p_\sigma(x)\|_2 \leq \beta(1 + \|x\|_2)$ for all $x$. Defining $A_\beta(x) \triangleq \frac{L}{m}\beta(1 + \|x\|_2)$, we note that, since $\nabla \log p = \frac{L}{m}\binom{L}{m}^{-1}\sum_{\sigma \in \binom{[L]}{m}} \nabla \log p_\sigma$, an application of the triangle inequality yields $\|\nabla \log p(x)\|_2 \leq A_\beta(x)$ for all $x$. Moreover, since $L/m \geq 1$ we have $\|\nabla \log p_\sigma(x)\|_2 \leq A_\beta(x)$ for all $x$ and $\sigma$.

From the construction in [22, Proof of Lem. 12], there is a random variable $Y$ with finite first moment such that the function $\tilde{g}_{h,\rho,\epsilon}(x) \triangleq \mathbb{E}[g_{h,\rho,\epsilon}(x - \epsilon Y)]$ satisfies

$$\|\mathcal{T}_P \tilde{g}_{h,\rho,\epsilon} - \mathcal{T}_P g_{h,\rho,\epsilon}\|_\infty \leq C_\rho \epsilon \tag{14}$$

and $\tilde{g}_{h,\rho,\epsilon} \in C_{\epsilon,\rho}\mathcal{G}_n$ for constants $C_\rho$ independent of $\epsilon$ and $h$ and $C_{\epsilon,\rho}$ independent of $h$.

**Showing the smoothed truncation $\tilde{g}_{h,\rho,\epsilon}$ is in a scaled copy of $\mathcal{G}_{b,n}$**   By Lemma 9, we have

$$\|A_\beta \tilde{g}_{h,\rho,\epsilon}\|_\infty \leq \|\Xi g_{h,\rho,\epsilon}\|_\infty \sqrt{2}\mathbb{E}[A_\beta(Y)] \leq \widetilde{\mathcal{M}}_P \triangleq \mathcal{M}'_P \sqrt{2}\mathbb{E}[A_\beta(Y)],$$

where $\widetilde{\mathcal{M}}_P$ is independent of $h, \epsilon$, and $\rho$. Thus for any $\sigma$, Cauchy-Schwarz, our bound (12), the triangle inequality, and the fact that $\|\nabla \log p_\sigma/A_\beta\|_\infty \leq 1$ and $\|\nabla \log p/A_\beta\|_\infty \leq 1$ imply

$$
\begin{aligned}
\|\tfrac{L}{m}\mathcal{T}_\sigma \tilde{g}_{h,\rho,\epsilon} - \mathcal{T}_P \tilde{g}_{h,\rho,\epsilon}\|_\infty &= \|\langle \tfrac{L}{m}\nabla \log p_\sigma - \nabla \log p, \tilde{g}_{h,\rho,\epsilon}\rangle\|_\infty \\
&\leq \|(\tfrac{L}{m}\nabla \log p_\sigma - \nabla \log p)/A_\beta\|_\infty \|A_\beta \tilde{g}_{h,\rho,\epsilon}\|_\infty \\
&\leq \widetilde{\mathcal{M}}_P(\tfrac{L}{m}\|\nabla \log p_\sigma/A_\beta\|_\infty + \|\nabla \log p/A_\beta\|_\infty) \leq (\tfrac{L}{m}+1)\widetilde{\mathcal{M}}_P.
\end{aligned}
$$

Thus, the triangle inequality and our error bounds (11), (13) and (14) yield

$$
\begin{aligned}
\|\mathcal{T}_P \tilde{g}_{h,\rho,\epsilon}\|_\infty &\leq \|\mathcal{T}_P g_h - \mathcal{T}_P g_{h,\rho}\|_\infty + \|\mathcal{T}_P g_{h,\rho} - \mathcal{T}_P g_{h,\rho,\epsilon}\|_\infty + \|\mathcal{T}_P g_{h,\rho,\epsilon} - \mathcal{T}_P \tilde{g}_{h,\rho,\epsilon}\|_\infty + \|\mathcal{T}_P g_h\|_\infty \\
&\leq C_1 \rho + C_0 + C_\rho \epsilon + 2 \quad \text{and} \\
\|\mathcal{T}_\sigma \tilde{g}_{h,\rho,\epsilon}\|_\infty &\leq \|\mathcal{T}_\sigma \tilde{g}_{h,\rho,\epsilon} - \tfrac{m}{L}\mathcal{T}_P \tilde{g}_{h,\rho,\epsilon}\|_\infty + \tfrac{m}{L}\|\mathcal{T}_P \tilde{g}_{h,\rho,\epsilon}\|_\infty \\
&\leq \tilde{C}_{\epsilon,\rho} \triangleq (1+\tfrac{m}{L})\widetilde{\mathcal{M}}_P + \tfrac{m}{L}(C_1 \rho + C_0 + C_\rho \epsilon + 2)
\end{aligned}
$$

for each $\sigma$. Therefore, $\tilde{g}_{h,\rho,\epsilon} \in \max(C_{\epsilon,\rho}, \tilde{C}_{\epsilon,\rho})\mathcal{G}_{b,n}$.

**Upper bounding the IPM $d_\mathcal{H}$**   Finally, we combine the triangle inequality and our approximation bounds (11), (13) and (14) once more to conclude

$$
\begin{aligned}
d_\mathcal{H}(Q_n, P) &\triangleq \sup_{h \in \mathcal{H}} |Q_n(h) - P(h)| = \sup_{h \in \mathcal{H}} |Q_n(\mathcal{T}_P g_h)| \\
&\leq \sup_{h \in \mathcal{H}} |Q_n(\mathcal{T}_P \tilde{g}_{h,\rho,\epsilon})| + |Q_n(\mathcal{T}_P \tilde{g}_{h,\rho,\epsilon} - \mathcal{T}_P g_{h,\rho,\epsilon})| + |Q_n(\mathcal{T}_P g_{h,\rho} - \mathcal{T}_P g_{h,\rho,\epsilon})| + |Q_n(\mathcal{T}_P g_h - \mathcal{T}_P g_{h,\rho})| \\
&\leq \sup_{h \in \mathcal{H}} |Q_n(\mathcal{T}_P \tilde{g}_{h,\rho,\epsilon})| + C_\rho \epsilon + C_0 Q_n(K_\epsilon^c) + C_1 \rho \\
&\leq \max(C_{\epsilon,\rho}, \tilde{C}_{\epsilon,\rho})\mathcal{S}(Q_n, \mathcal{T}_P, \mathcal{G}_{b,n}) + (C_0 + C_\rho)\epsilon + C_1 \rho.
\end{aligned}
$$

Since $\epsilon$ and $\rho$ were arbitrary, whenever $\mathcal{S}(Q_n, \mathcal{T}_P, \mathcal{G}_{b,n}) \to 0$, we have $d_\mathcal{H}(Q_n, P) \to 0$ and hence $Q_n \Rightarrow P$.

### C.2   Classical Stein set

Suppose $\mathcal{G}_n$ satisfies (A.2), and consider $\mathcal{G}_{k,\|\cdot\|_2}$ for $k(x,y) = \Phi(x-y) \triangleq (1+\|\Gamma(x-y)\|_2^2)^\beta$ with $\beta < 0$ and $\Gamma \succ 0$. Since $\nabla^s \Phi(0)$ is bounded for $s \in \{0,2,4\}$, [47, Cor. 4.36] implies that $\mathcal{G}_{k,\|\cdot\|_2} \subseteq c_0 \mathcal{G}_n$ for some $c_0$. The result now follows since $\mathcal{G}_{k,\|\cdot\|_2}$ also satisfies (A.1).

### C.3   Graph Stein set

If $\mathcal{G}_n$ satisfies (A.3), the result follows as $\mathcal{G}_n$ contains the classical Stein set $\mathcal{G}_{\|\cdot\|}$.

## D   Proof of Theorem 4: SSDs detect bounded SD non-convergence

Since $\mathcal{S}(Q_n, \mathcal{T}, \mathcal{G}_{b,n}) \not\to 0$, there exists $\epsilon > 0$ such that $\mathcal{S}(Q_n, \mathcal{T}, \mathcal{G}_{b,n}) > \epsilon$ infinitely often (i.o.). Fix any such $\epsilon$. For each $n$, choose $h_n = \mathcal{T}_P g_n$ for $g_n \in \mathcal{G}_{b,n}$ satisfying $Q_n(h_n) \geq \mathcal{S}(Q_n, \mathcal{T}, \mathcal{G}_{b,n}) - \epsilon/2$. Then since $\mathcal{T} = \binom{L}{m}^{-1} \tfrac{L}{m} \sum_{\sigma \in \binom{[L]}{m}} \mathcal{T}_\sigma$,

$$
\begin{aligned}
\mathcal{S}(Q_n, \mathcal{T}, \mathcal{G}_{b,n}) - \epsilon/2 &\leq Q_n(h_n) - \binom{L}{m}^{-1} \sum_{\sigma \in \binom{[L]}{m}} \mu_{n\sigma}(\mathcal{T}_\sigma g_n) + \binom{L}{m}^{-1} \sum_{\sigma \in \binom{[L]}{m}} \mu_{n\sigma}(\mathcal{T}_\sigma g_n) \\
&\leq \binom{L}{m}^{-1} \sum_{\sigma \in \binom{[L]}{m}} (\tfrac{L}{m}Q_n(\mathcal{T}_\sigma g_n) - \mu_{n\sigma}(\mathcal{T}_\sigma g_n)) + \mathcal{SS}(Q_n, \mathcal{T}, \mathcal{G}).
\end{aligned}
$$

Moreover, since $\|\mathcal{T}_\sigma g_n\|_\infty \leq 1$ for all $\sigma \in \binom{[L]}{m}$ and $n$, Lemma 10, proved in App. D.1, implies that $\tfrac{L}{m}Q_n(\mathcal{T}_\sigma g_n) - \mu_{n\sigma}(\mathcal{T}_\sigma g_n) \overset{a.s.}{\to} 0$ for each $\sigma$.

**Lemma 10** (Bounded function convergence). *Fix any triangular array of points $(x_i^n)_{i\in[n],n\geq 1}$ in $\mathbb{R}^d$, and, for each $n\geq 1$, define the measures*

$$\nu_n = \tfrac{1}{n}\sum_{i=1}^n \delta_{x_i^n} \quad and \quad \tilde{\nu}_n = \tfrac{1}{n}\sum_{i=1}^n \tfrac{B_i}{\tau}\delta_{x_i^n}$$

*where $B_i \overset{i.i.d.}{\sim} \text{Ber}(\tau)$ are independent Bernoulli random variables with $\mathbb{P}(B_i = 1) = \tau$. If $\|h_n\|_\infty \leq 1$ for each $n$, then, with probability 1,*

$$|\tilde{\nu}_n(h_n) - \nu_n(h_n)| \leq \tau^{-1}\sqrt{\tfrac{\log(n)+2\log(\log(n)))}{2n}}$$

*for all $n$ sufficiently large. Hence, $\tilde{\nu}_n(h_n) - \nu_n(h_n) \overset{a.s.}{\to} 0$.*

Hence

$$\mathbb{P}(\mathcal{SS}(Q_n,\mathcal{T},\mathcal{G}_n) \not\to 0) \geq \mathbb{P}(\mathcal{SS}(Q_n,\mathcal{T},\mathcal{G}_n) > \epsilon/2 \text{ i.o.})$$
$$\geq \mathbb{P}(Q_n(\mathcal{T}_\sigma g_n) - \mu_{n\sigma}(\mathcal{T}_\sigma g_n) < \tfrac{\epsilon}{2} \text{ eventually}, \forall\sigma) = 1$$

as advertised.

## D.1 Proof of Lemma 10: Bounded function convergence

The result will follow from the following lemma which establishes rates of convergence for subsampled measure expectations to their non-subsampled counterparts.

**Lemma 11.** *Under the notation of Lemma 10, for any $a \in [1,2]$, $\delta \in (0,1)$, and $h : \mathbb{R}^d \to \mathbb{R}$,*

$$\tilde{\nu}_n(h) - \nu_n(h) \leq \tfrac{\tau^{-1}\sqrt{\frac{1}{2}\log(1/\delta)}}{n^{1-1/a}}(\nu_n(|h|^a))^{1/a} \quad \text{with probability at least} \quad 1-\delta \quad and$$

$$\nu_n(h) - \tilde{\nu}_n(h) \leq \tfrac{\tau^{-1}\sqrt{\frac{1}{2}\log(1/\delta)}}{n^{1-1/a}}(\nu_n(|h|^a))^{1/a} \quad \text{with probability at least} \quad 1-\delta.$$

**Proof** Fix any $a \in [1,2]$, $\delta \in (0,1)$, and $h : \mathbb{R}^d \to \mathbb{R}$. Since

$$\tilde{\nu}_n(h) = \tfrac{1}{n}\sum_{i=1}^n \tfrac{B_i}{\tau}h(x_i^n)$$

is an average of independent variables $\tau^{-1}B_i h(x_i^n) \in \{0, \tau^{-1}h(x_i^n)\}$ with $\mathbb{E}[\tilde{\nu}_n(h)] = \nu_n(h)$, Hoeffding's inequality [26, Thm. 2] implies

$$\tilde{\nu}_n(h) - \nu_n(h) \leq \tau^{-1}\sqrt{\log(1/\delta)\tfrac{1}{2n^2}\sum_{i=1}^n h(x_i^n)^2} \quad \text{with probability at least} \quad 1-\delta \quad and$$

$$\nu_n(h) - \tilde{\nu}_n(h) \leq \tau^{-1}\sqrt{\log(1/\delta)\tfrac{1}{2n^2}\sum_{i=1}^n h(x_i^n)^2} \quad \text{with probability at least} \quad 1-\delta.$$

Moreover, since $\|\cdot\|_2 \leq \|\cdot\|_a$, we have $\sqrt{\sum_{i=1}^n h(x_i^n)^2/n^2} \leq (\sum_{i=1}^n |h(x_i^n)|^a/n^a)^{1/a}$, and the advertised result follows. $\square$

By Lemma 11 with $a = 2$,

$$\sum_{n=1}^\infty \mathbb{P}(|\nu_n(h_n) - \tilde{\nu}_n(h_n)| \geq \tau^{-1}\sqrt{\tfrac{\log(1/\delta_n)}{2n}}) \leq \sum_{n=1}^\infty \delta_n < \infty$$

for $\delta_n = 1/(n\log^2(n))$. The result now follows from the Borel-Cantelli lemma.

## E  Proof of Prop. 5: Coercive SSDs enforce tightness

Let $f(x) = \min_{\sigma\in\binom{[L]}{m}}\tfrac{L}{m}(\mathcal{T}_\sigma g)(x)$. Since $f$ is bounded below, $C = \inf_{x\in\mathbb{R}^d}f(x)$ is finite. Define

$$\gamma(r) \triangleq \inf\{f(x) - C : \|x\|_2 \geq r\},$$

so that $\gamma$ is nonnegative, coercive, and non-decreasing, as $f$ is coercive. Since $(Q_n)_{n=1}^\infty$ is not tight, there exist $\epsilon > 0$ and $R > 0$ such that $\limsup_n Q_n(\|X\|_2 > R) \geq \epsilon$ and $\gamma(R)\epsilon + C > 0$. Moreover, since $\gamma$ is non-decreasing and nonnegative, Markov's inequality gives

$$Q_n(\|X\|_2 > R) \leq Q_n(\gamma(\|X\|_2) > \gamma(R)) \leq \mathbb{E}_{Q_n}[\gamma(\|X\|_2)]/\gamma(R) \leq (Q_n(f) - C)/\gamma(R).$$

Meanwhile, our assumption on $g$ and the SSD subset representation (4) imply that, surely,

$$Q_n(f) = \tfrac{1}{n}\sum_{i=1}^n f(x_i) \leq \tfrac{1}{n}\sum_{i=1}^n \tfrac{L}{m}(\mathcal{T}_{\sigma_i}g)(x_i) \leq \mathcal{SS}(Q_n,\mathcal{T},\mathcal{G}_n).$$

Hence, $\mathcal{SS}(Q_n,\mathcal{T},\mathcal{G}_n)$ surely does not converge to zero, as

$$\limsup_n \mathcal{SS}(Q_n,\mathcal{T},\mathcal{G}_n) \geq \gamma(R)\limsup_n Q_n(\|X\|_2 > R) + C \geq \gamma(R)\epsilon + C > 0.$$

# F   Proof of Theorem 6: Coercive SSDs detect non-convergence

We consider each Stein set candidate in turn.

**Kernel Stein set**   Suppose $\mathcal{G}_n$ satisfies (A.1) for one of the specified kernels, $k_1(x,y) = \Phi_1(x-y)$ or $k_2(x,y) = \Phi_2(x-y)$, with $\Gamma = I_d$.

We have $\hat{\Phi}_1$ and $\hat{\Phi}_2$ are non-vanishing by [51, Thm. 8.15] and [9, Lem. 7], respectively. Moreover, we have for all $x,y \in \mathbb{R}^d$

$$\langle \nabla \log p(x) - \nabla \log p(y), x - y \rangle = \tfrac{L}{m} \binom{L}{m}^{-1} \sum_\sigma \langle \nabla \log p_\sigma(x) - \nabla \log p_\sigma(y), x - y \rangle$$
$$\leq -\kappa \|x-y\|_2^2 + r.$$

Hence if $Q_n \not\Rightarrow P$, then, by Theorem 3, either $\mathcal{S}(Q_n, \mathcal{T}_P, \mathcal{G}_{b,n}) \not\to 0$ or $(Q_n)_{n=1}^\infty$ is not tight.

If $\mathcal{S}(Q_n, \mathcal{T}_P, \mathcal{G}_{b,n}) \not\to 0$, then, with probability 1, $\mathcal{SS}(Q_n, \mathcal{T}_P, \mathcal{G}_n) \not\to 0$ by Theorem 4.

Now suppose $(Q_n)_{n=1}^\infty$ is not tight, and fix any $\sigma \in \binom{[L]}{m}$. Consider first the kernel $k_1$. Since $\frac{L}{m}\nabla \log p_\sigma$ has at most linear growth and satisfies distant dissipativity, the proof of [22, Lem. 16] constructs a function $g \in \mathcal{G}_n$ that is independent of the choice of $\sigma$ and satisfies $\frac{L}{m}\mathcal{T}_\sigma g \geq f_\sigma$ for some coercive bounded-below $f_\sigma$. Similarly, the same conclusion holds for the kernel $k_2$ by the proof of [9, Thm. 3]. Since $\binom{[L]}{m}$ has finite cardinality, we have $\frac{L}{m}\mathcal{T}_\sigma g \geq f$ for a common coercive bounded-below function $f(x) \triangleq \min_\sigma f_\sigma(x)$. Therefore, surely, $\mathcal{SS}(Q_n, \mathcal{T}_P, \mathcal{G}_n) \not\to 0$ by Prop. 5.

To extend this result to any $\Gamma \succ 0$, fix some $\Gamma \succ 0$. For any distribution $P$ on $\mathbb{R}^d$, let us write $\Gamma^{-1}P$ to represent the distribution of $\Gamma^{-1}Z$ when $Z \sim P$. Let $p_\Gamma$ be the density $\Gamma^{-1}P$. Then $p_\Gamma(x) = \det(\Gamma)\nabla \log p(\Gamma x)$ and $\nabla \log p_\Gamma(x) = \Gamma \nabla \log p(\Gamma x)$, and for any $\sigma \in \binom{[L]}{m}$, the analog $p_{\Gamma,\sigma}$ of $p_\Gamma$ satisfies $p_{\Gamma,\sigma}(x) = \det(\Gamma)\nabla \log p_\sigma(\Gamma x)$ and $\nabla \log p_{\Gamma,\sigma}(x) = \Gamma \nabla \log p_\sigma(\Gamma x)$. By the same argument made in [10, Lem. 4], we have that $\nabla \log p_\Gamma$ is Lipschitz and $\nabla \log p_{\Gamma,\sigma}$ satisfies distant dissipativity. And since

$$\frac{\|\nabla \log p_{\Gamma,\sigma}(x)\|_2}{1 + \|x\|_2} = \frac{\|\Gamma \nabla \log p_\sigma(\Gamma x)\|_2}{1 + \|\Gamma x\|_2}\frac{1 + \|\Gamma x\|_2}{1 + \|x\|_2} \leq \|\Gamma\|_{\mathrm{op}}(1 + \|\Gamma\|_{\mathrm{op}})\frac{\|\nabla \log p_\sigma(\Gamma x)\|_2}{1 + \|\Gamma x\|_2}$$

is uniformly bounded, we can apply the same argument discussed in [10, Lem. 4], i.e., make a global change of coordinates $x \mapsto \Gamma^{-1}x$ and then invoke Theorem 6 for $\Gamma^{-1}P$ and $\Gamma^{-1}Q_n$ with a non-preconditioned kernel, thereby concluding the proof.

**Classical Stein set**   Suppose $\mathcal{G}_n = \mathcal{G}_{\|\cdot\|}$ satisfies (A.2). By the proof of Theorem 3, for $\Gamma = I$ and any $\beta \in (-1,0)$, there is a constant $c_0 > 0$ such that the kernel Stein set $\mathcal{G}_{k,\|\cdot\|_2} \subseteq c_0\mathcal{G}_n$. Hence $\mathcal{SS}(Q_n, \mathcal{T}_P, \mathcal{G}_{k,\|\cdot\|_2}) \leq c_0\mathcal{SS}(Q_n, \mathcal{T}_P, \mathcal{G}_n)$ for all $n$ implying the result.

**Graph Stein set**   Suppose $\mathcal{G}_n$ satisfies (A.3). Then the result follows as $\mathcal{G}_n$ contains the classical Stein set $\mathcal{G}_{\|\cdot\|}$.

# G   Proof of Theorem 7: Wasserstein convergence of SVGD and SSVGD

## G.1   Additional notation

For each $\epsilon > 0$ and collection of $n$ points $(x_i^n)_{i=1}^n$ with associated discrete measure $\nu_n = \frac{1}{n}\sum_{i=1}^n \delta_{x_i^n}$, we define the random one-step SSVGD mapping

$$T_{\nu_n,\epsilon,n}^m(x) = x + \epsilon \tfrac{1}{n}\sum_{j=1}^n \tfrac{L}{m}\nabla \log p_{\sigma_j}(x_j^n)k(x_j^n, x) + \nabla_{x_j^n}k(x_j^n, x)$$

for $(\sigma_j)_{j=1}^n$ independent uniformly random size-$m$ subsets of $[L]$. We also let $\Phi_{\epsilon,n}^m(\mu)$ denote the random distribution of $T_{\nu_n,\epsilon,n}^m(X)$ when $X \sim \mu$.

## G.2   Proof of Theorem 7

We will prove each convergence claim by induction on $r \geq 0$.

**Inductive proof of** $W_1(Q_{n,r}, Q_{\infty,r}) \to 0$   For our base case we have $W_1(Q_{n,0}, Q_{\infty,0}) \to 0$ by assumption.

Now, fix any $r \geq 0$ and assume $W_1(Q_{n,r}, Q_{\infty,r}) \to 0$, so that $c_0(1 + \|\cdot\|_2)$ is uniformly $Q_{n,r}$-integrable and $Q_{n,\infty}$-integrable by [17, Proof of Cor. 1]. Therefore, there exists a constant $C' > 0$ such that

$$\sup_{n \geq 1} 1 + \epsilon_r c_1(1 + Q_{n,r}(\|\cdot\|_2)) + \epsilon_r c_2(1 + Q_{\infty,r}(\|\cdot\|_2)) \leq C'.$$

Now, note that

$$W_1(Q_{n,r+1}, Q_{\infty,r+1}) = W_1(\Phi_{\epsilon_r}(Q_{n,r}), \Phi_{\epsilon_r}(Q_{\infty,r})).$$

To control this expression, we provide a lemma, proved in App. G.3, which establishes the pseudo-Lipschitzness of the one-step SVGD mapping $\Phi_\epsilon$.

**Lemma 12** (Wasserstein pseudo-Lipschitzness of SVGD). *Suppose that, for some $c_1, c_2 > 0$,*

$$\sup_{z \in \mathbb{R}^d} \|\nabla_z(\nabla \log p(x)k(x,z) + \nabla_x k(x,z))\|_{op} \leq c_1(1 + \|x\|_2) \quad and$$
$$\sup_{x \in \mathbb{R}^d} \|\nabla_x(\nabla \log p(x)k(x,z) + \nabla_x k(x,z))\|_{op} \leq c_2(1 + \|z\|_2).$$

*Then, for any $\epsilon > 0$ and probability measures $\mu, \nu$,*

$$W_1(\Phi_\epsilon(\mu), \Phi_\epsilon(\nu)) \leq W_1(\mu, \nu)(1 + \epsilon c_1(1 + \mu(\|\cdot\|_2)) + \epsilon c_2(1 + \nu(\|\cdot\|_2))).$$

Our pseudo-Lipschitz assumptions (7) and Lemma 12 imply

$$W_1(\Phi_{\epsilon_r}(Q_{n,r}), \Phi_{\epsilon_r}(Q_{\infty,r})) \leq W_1(Q_{n,r}, Q_{\infty,r})(1 + \epsilon_r c_1(1 + Q_{n,r}(\|\cdot\|_2)) + \epsilon_r c_2(1 + Q_{\infty,r}(\|\cdot\|_2)))$$
$$\leq C' W_1(Q_{n,r}, Q_{\infty,r}) \to 0,$$

proving our first claim.

**Inductive proof of** $W_1(Q^m_{n,r}, Q_{n,r}) \to 0$   For our base case we have, $W_1(Q^m_{n,0}, Q_{n,0}) = 0$.

Now fix any $r \geq 0$, let $\mathcal{E}$ be the event on which $W_1(Q^m_{n,r}, Q_{n,r}) \to 0$ as $n \to \infty$, and assume $\mathbb{P}(\mathcal{E}) = 1$. Since $W_1(Q_{n,r}, Q_{\infty,r}) \to 0$, on $\mathcal{E}$ we find that $W_1(Q^m_{n,r}, Q_{\infty,r}) \to 0$ and hence $c_0(1 + \|\cdot\|_2)$ is uniformly $Q^m_{n,r}$-integrable and uniformly $Q_{n,r}$-integrable by [17, Proof of Cor. 1]. Therefore, on $\mathcal{E}$, there exists a constant $C$ such that

$$\sup_{n \geq 1} 1 + \epsilon_r c_1(1 + Q^m_{n,r}(\|\cdot\|_2)) + \epsilon_r c_2(1 + Q_{n,r}(\|\cdot\|_2)) \leq C.$$

By the triangle inequality,

$$W_1(Q^m_{n,r+1}, Q_{n,r+1}) = W_1(\Phi^m_{\epsilon_r,n}(Q^m_{n,r}), \Phi_{\epsilon_r}(Q_{n,r}))$$
$$\leq W_1(\Phi^m_{\epsilon_r,n}(Q^m_{n,r}), \Phi_{\epsilon_r}(Q^m_{n,r})) + W_1(\Phi_{\epsilon_r}(Q^m_{n,r}), \Phi_{\epsilon_r}(Q_{n,r})).$$

On $\mathcal{E}$, our growth assumptions (8), the uniformly $Q^m_{n,r}$-integrability of $c_0(1 + \|\cdot\|_2)$, and the following lemma, proved in App. G.4, establish that the Wasserstein distance $W_1(\Phi^m_{\epsilon_r,n}(Q^m_{n,r}), \Phi_{\epsilon_r}(Q^m_{n,r}))$ between one step of SSVGD and one step of SVGD from a common starting point converges to 0 almost surely as $n$ grows.

**Lemma 13** (One-step convergence of SSVGD to SVGD). *Fix any triangular array of points $(x^n_i)_{i \in [n], n \geq 1}$ in $\mathbb{R}^d$, and define the discrete probability measures $\nu_n = \frac{1}{n} \sum_{i=1}^n \delta_{x^n_i}$. Suppose $\nabla \log p_\sigma(\cdot)k(\cdot, z)$ is continuous for each $z \in \mathbb{R}^d$ and $\sigma \in \binom{[L]}{m}$ and let*

$$f_0(x) \triangleq \sup_{z \in \mathbb{R}^d, \sigma \in \binom{[L]}{m}} \|\nabla \log p_\sigma(x)\|_\infty |k(x,z)|,$$
$$f_1(x) \triangleq \sup_{z \in \mathbb{R}^d, \sigma \in \binom{[L]}{m}} \|\nabla_x(\nabla \log p_\sigma(x)k(x,z))\|_{op}.$$

*If $f_0$ is $\nu_n$-uniformly integrable and $f_0, f_1$ are bounded on each compact set, then, for any $\epsilon > 0$, $W_1(\Phi^m_{\epsilon,n}(\nu_n), \Phi_\epsilon(\nu_n)) \overset{a.s.}{\to} 0$ as $n \to \infty$.*

In addition, on $\mathcal{E}$, our pseudo-Lipschitz assumptions (7) and Lemma 12 imply

$$W_1(\Phi_{\epsilon_r}(Q^m_{n,r}), \Phi_{\epsilon_r}(Q_{n,r})) \leq W_1(Q^m_{n,r}, Q_{n,r})(1 + \epsilon c_1(1 + Q^m_{n,r}(\|\cdot\|_2)) + \epsilon c_2(1 + Q_{n,r}(\|\cdot\|_2)))$$
$$\leq C W_1(Q^m_{n,r}, Q_{n,r}) \to 0.$$

Hence, on $\mathcal{E}$, $W_1(Q^m_{n,r+1}, Q_{n,r+1}) \overset{a.s.}{\to} 0$, proving our second claim.

### G.3 Proof of Lemma 12: Wasserstein pseudo-Lipschitzness of SVGD

Assume that $\mu$ and $\nu$ have integrable means (or else the advertised claim is vacuous), and select $(X', Z')$ to be an optimal 1-Wasserstein coupling of $(\mu, \nu)$. The triangle inequality, Jensen's inequality, and our pseudo-Lipschitzness assumptions imply that

$$
\begin{aligned}
&\|T_{\mu,\epsilon}(x) - T_{\nu,\epsilon}(z)\|_2 \\
&\leq \|x - z\|_2 \\
&\quad + \epsilon\|\mathbb{E}[\nabla \log p(X')k(X', x) + \nabla_{x'}k(X', x) - (\nabla \log p(X')k(X', z) + \nabla k(X', z))]\|_2 \\
&\quad + \epsilon\|\mathbb{E}[\nabla \log p(X')k(X', z) + \nabla_{x'}k(X', z) - (\nabla \log p(Z')k(Z', z) + \nabla_{z'}k(Z', z))]\|_2 \\
&\leq \|x - z\|_2(1 + \epsilon c_1(1 + \mathbb{E}[\|X'\|_2])) + \epsilon c_2 \mathbb{E}[\|X' - Z'\|_2](1 + \|z\|_2) \\
&= \|x - z\|_2(1 + \epsilon c_1(1 + \mu(\|\cdot\|_2))) + \epsilon c_2 W_1(\mu, \nu)(1 + \|z\|_2).
\end{aligned}
$$

Since $T_{\mu,\epsilon}(X') \sim \Phi_\epsilon(\mu)$ and $T_{\nu,\epsilon}(Z') \sim \Phi_\epsilon(\nu)$, we conclude that

$$
\begin{aligned}
W_1(\Phi_\epsilon(\mu), \Phi_\epsilon(\nu)) &\leq \mathbb{E}[\|T_{\mu,\epsilon}(X') - T_{\nu,\epsilon}(Z')\|_2] \\
&\leq \mathbb{E}[\|X' - Z'\|_2](1 + \epsilon c_1(1 + \mu(\|\cdot\|_2))) + \epsilon c_2 W_1(\mu, \nu)(1 + \mathbb{E}[\|Z'\|_2]) \\
&= W_1(\mu, \nu)(1 + \epsilon c_1(1 + \mu(\|\cdot\|_2)) + \epsilon c_2(1 + \nu(\|\cdot\|_2))).
\end{aligned}
$$

### G.4 Proof of Lemma 13: One-step convergence of SSVGD to SVGD

Note that the random one-step SSVGD mapping takes the form

$$
T^m_{\nu_n, \epsilon, n}(x) = x + \epsilon \nu_n(\nabla_{x^n_j} k(\cdot, x)) + \epsilon \binom{L}{m}^{-1} \sum_{\sigma \in \binom{[L]}{m}} \nu_{n\sigma}(\nabla \log p_\sigma(\cdot)k(\cdot, x))
$$

for $\nu_{n\sigma} = \binom{L}{m} \frac{L}{m} \frac{1}{n} \sum_{j=1}^n B_{j\sigma} \delta_{x^n_j}$ and $B_{j\sigma} = \mathbb{I}[\sigma = \sigma_j]$. Moreover, by Kantorovich-Rubinstein duality, we may write the 1-Wasserstein distance as

$$
\begin{aligned}
&W_1(\Phi^m_{\epsilon, n}(\nu_n), \Phi_\epsilon(\nu_n)) \\
&= \sup_{f:M_1(f)\leq 1} \Phi^m_{\epsilon, n}(\nu_n)(f) - \Phi_\epsilon(\nu_n)(f) \\
&= \sup_{f:M_1(f)\leq 1} \frac{1}{n} \sum_{i=1}^n f(T^m_{\nu_n, \epsilon, n}(x^n_i)) - f(T_{\nu_n, \epsilon}(x^n_i)) \\
&\leq \frac{1}{n} \sum_{i=1}^n \|T^m_{\nu_n, \epsilon, n}(x^n_i) - T_{\nu_n, \epsilon}(x^n_i)\|_2 \\
&= \binom{L}{m}^{-1} \frac{\epsilon}{n} \sum_{i=1}^n \|\sum_\sigma \frac{L}{m} \nu_n(\nabla \log p_\sigma(\cdot)k(\cdot, x^n_i)) - \nu_{n\sigma}(\nabla \log p_\sigma(\cdot)k(\cdot, x^n_i))\|_2 \\
&\leq \binom{L}{m}^{-1} \sum_\sigma \frac{\epsilon\sqrt{d}}{n} \sum_{i=1}^n \|\frac{L}{m}\nu_n(\nabla \log p_\sigma(\cdot)k(\cdot, x^n_i)) - \nu_{n\sigma}(\nabla \log p_\sigma(\cdot)k(\cdot, x^n_i))\|_\infty \\
&\leq \epsilon\sqrt{d}\binom{L}{m}^{-1} \sum_\sigma \sup_{h\in\mathcal{H}_f} |\nu_{n\sigma}(h) - \frac{L}{m}\nu_n(h)|. \quad (15)
\end{aligned}
$$

where we have used the triangle inequality and norm relation $\|\cdot\|_2 \leq \sqrt{d}\|\cdot\|_\infty$ in the penultimate display and $\mathcal{H}_f$ is defined in the statement of Lemma 8.

For each $\sigma \in \binom{[L]}{m}$, since $|f_0|$ is uniformly $\nu_n$-integrable, and $\nu_{n\sigma}(|f_0|I_K) \leq \binom{L}{m}\frac{L}{m}\nu_n(|f_0|I_K)$ for every compact set $K$, we find that $|f_0|$ is uniformly $\nu_{n\sigma}$-integrable for each $\sigma$. Letting $I_{B_R}(x) = \mathbb{I}[\|x\|_2 \leq R]$, for each $\sigma$, since $\nu_{n\sigma}(hI_{B_R}) - \frac{L}{m}\nu_n(hI_{B_R}) \overset{a.s.}{\to} 0$ for any $R > 0$ and any bounded $h$ by Lemma 10, we have $\sup_{h\in\mathcal{H}_f} |\nu_{n\sigma}(h) - \frac{L}{m}\nu_n(h)| \overset{a.s.}{\to} 0$ by Lemma 8. The result now follows from the bound (15).