[Reviews · NeurIPS 2020]

Review 1

Summary and Contributions: Update: I have read the rebuttal. I think the algorithmic contribution of this paper is limited, but I like the theoretical contribution more. Thus I incline to keep my score. ========= Stein discrepancy has been used in machine learning to measure distance between two distributions with a number of downstream applications such as in Bayesian inference. In practice, a stochastic version of the Stein discrepancy is typically used to derive algorithms for large datasets, without a formal theoretical justification. This paper fills this gap by proving theoretical justifications for the use of stochastic Stein discrepancy. The theory justify the reasonability of using SSDs for measuring convergence of two distributions and also for Bayesian inference. The main contribution of this paper lies in the theoretical side, as stochastic algorithms with Stein discrepancy have been widely used in practice.

Strengths: The theory is sound and seems correct. It proves the first guarantee for a practical use of SD.

Weaknesses: The results are not surprising, and have limited impacts in practice. The paper could be stronger if it can find out some distintive properties of SSD compared to SD.

Correctness: The claims, method and empirical results seem correct.

Clarity: The paper is written fairly well.

Relation to Prior Work: Yes

Reproducibility: Yes

Additional Feedback: I like the theoretical contribution of this paper, which fills the gap between theory and practical use of SD in previous literature. A downside of the theoretical results is that it is hard to see the advantage of SSD over SD, partly because the current theory is in terms of asymptotic convergence. I would recommend the authors the look at the non-asymptotic behavior of SSD with finite number of particles. I acknowledge that this could be challenging, as even the convergence of SD is hard to guarantee, except for a modified version by Zhang et al., 2020. There are some detailed comments: 1. It is better to define what "coercive" means in Proposition 5. 2. Theorem 6 seems to be limited to a special polynomial kernel. In practice, a Gaussian kernel is typically used. Does Theorem 6 applies to this case? 3. I don't understand Theorem 4. It is claimed before Theorem 4 that the theorem proves non-convergence property of SD in terms of if Q \not\to P, then S(...) \not\to 0. But Theorem 4 seems to prove S(...) \to 0. Am I missing something? Also, what is the point of introducing Theorem 4? It seems the theorem is about SD but not SSD. Zhang et al., stochastic particle-optimization sampling and the non-asymptotic convergence theory, 2020.


Review 2

Summary and Contributions: Post-rebuttal update: thank you for the response. I've updated my score given the updated proof. I still share the concern of R3 on the discrete case, and I agree with R1 that the results are to some extent unsurprising. ==== This work investigates a variant of Stein discrepancies where the target log density, log p, is replaced with its stochastic approximation. Such stochastic Stein discrepancies are shown to be able to detect commonly-used convergence and non-convergence properties. It also establishes convergence results for both SVGD and its stochastic variants under more realistic conditions, allowing the log density of the true posterior to have quadratic growth beyond a compact domain.

Strengths: The results are relevant since stochastic approximations of Stein discrepancies are widely used, yet previously the impact of stochastic approximation was not discussed. Moreover, improved analysis of SVGD might be of independent interest to the VI community.

Weaknesses: * In the context of posterior inference (i.e. p corresponds to a posterior), the stochastic Stein discrepancy investigated here is technically different from the common practice: here separate mini-batches of data points are drawn for each sample point, while in practice we usually use a fixed mini-batch for all sample points. The version discussed here is more difficult to implement. * I have concerns about the correctness of the proof; see below.

Correctness: I skimmed through Appendix A-D and didn't find any mistakes. Appendix E: where are the other two cases (A.2, A.3) discussed? Appendix F: The proof of Eq.(8) doesn't seem correct, specifically the last inequality below L512 doesn't follow from union bound.

Clarity: Yes.

Relation to Prior Work: Yes.

Reproducibility: Yes

Additional Feedback: Clarification of App.E-F would be most appreciated.


Review 3

Summary and Contributions: The Stein discrepancy (SD) serves as a measure between distributions for approximate inference while the exact computation of SD can be prohibitive. To mitigate the computation cost, the authors propose the stochastic Stein discrepancy that is easier to compute and serves as an estimate of the SD by subsampling the Stein operator. For a decomposable Stein operator, the coordinate computation for SSD involves only a subset of the Stein operator instead of the whole Stein operator as SD and therefore results in less computation cost. The authors also provide theoretical analysis showing that SSD has the same nice convergence properties as SD, that is, SSD can detect the convergence/non-convergence almost surely. Experiments show that SSD has similar performance to SD in various inference tasks.

Strengths: The SSD can be potentially useful for scaling up a bunch of SD-based inference methods since the decomposable Stein operators are widely used. It provides a tradeoff between efficiency and accuracy in estimating SD. Also, the SSD comes with convergence guarantees under some assumptions showing that the convergence properties of SSD are as nice as the corresponding SD.

Weaknesses: I found the word subsampling confusing because it usually refers to subsampling from the points. While the major computation bottleneck of SD is to compute the gradients of all $n$ points and this is especially true when the number of points is large, SSD still requires such computations over all $n$ points. What SSD reduces is the complexity term in the computation of each gradient, from $L$ the total number of base operators to $m$ the subset size of the base operators. I would suggest adding some comparison between subsampling points and subsampling the base operators to avoid confusion.

Correctness: Although theoretical analysis make sense to me, some missing definitions for nations make the paper less readable: line 111: p_{sigma_i} is not defined. It seems that this should be defined as a summation with index running over set sigma_i in similar way to the operator T_sigma. Algorithm 1: R is not defined. line 207: Q_n^m is not defined.

Clarity: The paper is generally well written. Still, there are some minor issues besides the missing definitions mentioned above. line 59: "discrete" is mentioned. However all the analysis on the paper are based on that the distribution being continuous. line 60: "will an employ" -> "will employ"

Relation to Prior Work: Previous work [1,2] are closely related where SD with coordinate-wise kernels is proposed to approximate SD. They subsampled base operators for each coordinate while SSD does so for each point. [1] Wang, D., Zeng, Z., and Liu, Q. Stein variational message passing for continuous graphical models.InInternational Conference on Machine Learning, pp. 5206–5214, 2018. [2] Zhuo, J., Liu, C., Shi, J., Zhu, J., Chen, N., and Zhang, B. Message passing Stein variational gradient descent. InInternational Conference on Machine Learning, pp. 6013–6022, 2018.11

Reproducibility: Yes

Additional Feedback:


Review 4

Summary and Contributions: The authors study the theoretical convergence properties of stochastic Stein discrepancies based on subsampling proposed in other papers, and show that SSDs achieves the same convergence as standard SDs with probability 1. Moreover, they analyze the convergence properties for SVGD and its stochastic version by extending the results in [30] for SVGD from bounded domains to unbounded domains.

Strengths: The strengths of the work are: 1. rigorous analysis of the detect convergence and detect non-convergence of SSDs. 2. extension of the convergence analysis for SVGD in [30] to more general cases that hold for the commonly used kernels. 3. interesting and convincing demonstration by empirical examples.

Weaknesses: I do not see evident weaknesses of this work. The proof in the continuous case does not seem to be directly applicable to the discrete case.

Correctness: The convergence results, methods, and empirical methodology seem to be correct, though I did not check the proofs.

Clarity: The paper is well written and clear to follow.

Relation to Prior Work: The discussion of the difference between this work and prior contributions is clear.

Reproducibility: Yes

Additional Feedback:

[Author Response · NeurIPS 2020]

We thank all reviewers for their encouraging and constructive feedback and respond to each in turn.

**(R1) SSD advantages:** The chief advantages of SSDs are that they require orders of magnitude less computation than
SDs (while still determining convergence), can be deployed when exact SDs are simply infeasible (e.g., in the settings
motivating many approximate MCMC methods), and, for a fixed computational budget, typically yield more accurate
posterior approximations than exact SVGD.

**(R1) Gaussian kernel:** Thm. 6 of [22] showed that (non-stochastic) KSDs based on the Gaussian kernel fail to detect
non-convergence (and thus often have terrible power in practice) even for simple target distributions like multivariate
Gaussians; for the same reason SSDs based on the same kernel fail to detect non-convergence. We focused on the
inverse multiquadric kernel (which has very different properties from a polynomial kernel), because its KSD detects
non-convergence in great generality. Our results apply to other kernels, such as the inverse log kernel of [9, Thm. 3].

**(R1) Thm. 4:** We apologize for the confusion: the text preceding Thm. 4 is the contrapositive of (and hence equivalent
to) the statement in Thm. 4. We will reword Thm. 4 to improve clarity and note here that Thm. 4 does not prove that
$\mathcal{S}(\nu_n) \to 0$ but rather that $\nu_n \Rightarrow P$ if $\mathcal{S}(\nu_n) \to 0$ and $\nu_n$ is tight. We will also provide a roadmap at the start of Sec.
4.2 to clarify how the results fit together: we show that SSDs detect non-convergence (Thm. 6) in a series of steps: (a)
by Thm. 4, if $Q_n \not\Rightarrow P$ then either a bounded SD $\not\to 0$ or $Q_n$ is not tight; (b) by Thm. 3, if a bounded SD $\not\to 0$ then its
SSD $\not\to 0$ w.p. 1; (c) by Prop. 5, if $Q_n$ is not tight, then the SSD $\not\to 0$ surely. Here, the new result on bounded SD
non-convergence (Thm. 4) is an important stepping stone to establishing SSD non-convergence (Thm. 6).

**(R2) Minibatches:** In the revision, we will clarify that a separate minibatch per sample point is standard in the SD
context: it is used in the original SVGD paper [32] and each of the cited uses of SSDs [2, 40]. A separate minibatch per
sample point is also standard in each of the approximate MCMC algorithms discussed [8, 14], including stochastic
gradient Langevin dynamics [48] and SGFS [1]. We will also highlight the substantial advantage of using separate
minibatches over a single minibatch. If $P$ is the target and $\tilde{P}$ is the posterior induced by a single minibatch of data, then
the separate minibatch SSD is guaranteed to detect convergence and non-convergence to $P$ for any minibatch size (by
our Thms. 2 & 6), but a single minibatch SSD cannot correctly discriminate between $P$ and $\tilde{P}$ (it will incorrectly
declare that samples from $\tilde{P}$ are converging to $P$ and incorrectly declare that samples from $P$ are not converging to $P$).

**(R2) App. E:** Thank you for pointing out this inadvertent omission. The revision will reflect that, exactly as in the
proof of Thm. 4, the other two cases follow as their Stein sets contain a scaled copy of the kernel Stein set.

**(R2) App. F:** Thank you for flagging this error. We have corrected the statement using $\mathcal{H} = \{h : \|h\|_\infty + \mathrm{Lip}(h) < 1\}$:
**Lemma 1.** *If two sequences of random measures $(\nu_n)_{n=1}^\infty$ and $(\tilde{\nu}_n)_{n=1}^\infty$ on $\mathbb{R}^d$ satisfy $\nu_n(hI_{B_R}) - \tilde{\nu}_n(hI_{B_R}) \overset{a.s.}{\to} 0$ for*
*each $h \in C_b$ and some $B_R \triangleq \{\|x\|_2 \le R\}$ with $R \ge S$ for all $S > 0$, then $\sup_{h \in \mathcal{H}} |\nu_n(hI_{B_R}) - \tilde{\nu}_n(hI_{B_R})| \overset{a.s.}{\to} 0$ for*
*each $R > 0$. If, in addition, $f_0$ is almost surely uniformly $\nu_n$-integrable and uniformly $\tilde{\nu}_n$-integrable, and $f_0, f_1$ are*
*bounded on compact sets, then $\sup_{h \in \mathcal{H}_f} |\nu_n(h) - \tilde{\nu}_n(h)| \overset{a.s.}{\to} 0$ for $\mathcal{H}_f = \{h : |h| \le f_0, \frac{|h(x)-h(y)|}{\|x-y\|_2} \le f_1(x), \forall x, y\}$.*
**Proof** Fix $R, \epsilon > 0$, and let $K = B_R$. By the Arzelà–Ascoli theorem, there exists a finite $\epsilon/2$ subcover of the set of
$K$-restrictions $\{h|_K : h \in \mathcal{H}\}$, extendable to $C_b$ functions $(h_k)_{k=1}^m$ on $\mathbb{R}^d$. The union bound and our assumption now
give $\mathbb{P}(\sup_{h \in \mathcal{H}} |\nu_n(hI_K) - \tilde{\nu}_n(hI_K)| > \epsilon \text{ i.o.}) \le \mathbb{P}(\max_{1 \le k \le m} |\nu_n(h_kI_K) - \tilde{\nu}_n(h_kI_K)| > \epsilon/2 \text{ i.o.})$
$\le \sum_{k=1}^m \mathbb{P}(|\nu_n(h_kI_K) - \tilde{\nu}_n(h_kI_K)| > \epsilon/2 \text{ i.o.}) = 0$. As $\mathcal{H}_f$ is uniformly bounded-Lipschitz on $B_R$, the second claim
follows as in the submission with $\mathcal{H}_f$ and $\mathcal{H}$ replacing $C(\mathbb{R}^d) : |h| \le |f|$ and $C(\mathbb{R}^d)$ and $K_\epsilon = B_R$ for suitable $R$. $\square$
For any target $P$, there exists a sequence of radii $(R_j)_{j=1}^\infty$ with $R_j \to \infty$ such that $\mathbb{E}_P(\|X\|_2 = R_j) = 0$ so that $B_{R_j}$ is
a continuity set under $P$. Since $Q_n \Rightarrow P$, we have $Q_n(hI_{B_{R_j}}) \to P(hI_{B_{R_j}})$ for each $j$ and $h \in C_b$ by the Portmanteau
theorem. Thm. 2 now follows assuming $\sup_{g \in \mathcal{G}_n, y} \|\mathcal{T}_l g(x) - \mathcal{T}_l g(y)\|_2 / \|x - y\|_2$ bounded on compact sets (which
holds for the pairings in [21-23]), and Thm. 7 and Lem. 11 follow assuming $\sup_{l \in [L], z \in \mathbb{R}^d} \|\nabla_x(\nabla \log p_l(x)k(x,z))\|_2$
bounded on compact sets (which holds for $\nabla \log p_l$ in $C^1$ and bounded-Lipschitz $k$).

**(R3) Subsampling:** Prior work, like the finite set SD of [28] and the random feature SDs of [27] address the $n^2$
complexity of SDs by introducing alternative SDs with O(n) complexity. We will clarify that our work addresses the
complementary problem of an expensive Stein operator and should not be viewed as an alternative to O(n)-time SDs.
Rather, datapoint subsampling can be directly applied to O(n) SDs to obtain O(n) SSDs with additional speed-ups.

**(R3) Discrete:** In the revision, we will clarify that while we develop the most extensive theory for the popular Langevin
Stein operator, our results on detecting convergence (Thm. 2), enforcing tightness (Prop. 5), and detecting bounded
non-convergence (Thm. 3) apply to any Stein operator and to both discrete and continuous targets.

**(R3) Coordinate kernels:** Thank you for these references. In the revision, we will highlight that both works can be
viewed as deploying exact SDs with special Stein sets featuring coordinate-dependent kernels. Since every coordinate
is still updated on each SVGD step, this is somewhat different from, for example, subsampling coordinate operators for
computational benefit (in which case certain coordinates would not be updated at all on each SVGD step). However,
these SDs can be combined with datapoint subsampling to obtain substantial speed-ups.

[Meta-Review · NeurIPS 2020]

The reviewers agree that this paper has some interesting ideas and would be a useful contribution to NeurIPS. The reviewers had a few concerns though, please reread the reviews and try to address them in the final manuscript (especially concerning clarity, and maybe the addition of a short paragraph concerning the applicability of the proof to the discrete case).